# Differentiable Learning of Generalized Structured Matrices for Efficient Deep Neural Networks

**Changwoo Lee, Hun-Seok Kim**
University of Michigan, Ann Arbor, MI, USA
`{cwoolee, hunseok}@umich.edu`

## Abstract

This paper investigates efficient deep neural networks (DNNs) to replace dense unstructured weight matrices with structured ones that possess desired properties. The challenge arises because the optimal weight matrix structure in popular neural network models is obscure in most cases and may vary from layer to layer even in the same network. Prior structured matrices proposed for efficient DNNs were mostly hand-crafted without a generalized framework to systematically learn them. To address this issue, we propose a generalized and differentiable framework to learn efficient structures of weight matrices by gradient descent. We first define a new class of structured matrices that covers a wide range of structured matrices in the literature by adjusting the structural parameters. Then, the frequency-domain differentiable parameterization scheme based on the Gaussian-Dirichlet kernel is adopted to learn the structural parameters by proximal gradient descent. On the image and language tasks, our method learns efficient DNNs with structured matrices, achieving lower complexity and/or higher performance than prior approaches that employ low-rank, block-sparse, or block-low-rank matrices.

## 1 Introduction

Deep Neural Networks (DNNs) for large language models (LLMs) (Vaswani et al., 2017; Devlin et al., 2018; Radford et al., 2019; Brown et al., 2020) and vision tasks (Dosovitskiy et al., 2020; Touvron et al., 2021) have shown great success in various domains in recent years. The size of the DNNs, however, has increased on an extraordinary scale – up to 70 Billion parameters in the single model Zhao et al. (2023) – requiring an unprecedented amount of computing resources and energy to deploy the DNN-based services. Fortunately, under certain conditions, the weight matrices of DNNs trained by stochastic gradient descent (SGD) naturally have well-defined preferred structures, such as low-rank matrices (Yaras et al., 2023; Huh et al., 2022; Gunasekar et al., 2017). However, identifying such structures to lower the effective complexity of weight matrices in recent models such as Transformers Vaswani et al. (2017) remains a challenging problem. Also, it mostly relies on the existing human-designed / hand-crafted structured matrices without a unified systematic approach. Hence prior works have focused on investigating new classes of structured matrices for DNNs (Li et al., 2015; Dao et al., 2022; Chen et al., 2022). Notably, each structured matrix is defined disjointedly from other formats. For example, neither the block-sparse matrix format nor the low-rank matrix format of the same number of parameters is a subset of another, and yet there is no unified representation that describes both well. Moreover, the structure description and the implementation complexity of the structured matrix are in non-differentiable discrete spaces.

In this paper, we investigate (locally) optimal structures of the weight matrices as well as a differentiable training method to learn them, attempting to answer the following two questions:

1. *Is there a universal format that represents a wide range of structured matrices?*
2. *Can the structure of such matrices be learned efficiently, if it exists?*

**Contributions.** Tackling the above two questions, we introduce a *generalized* and *differentiable* structured matrix format. The main contributions of this work can be summarized as follows.

1) We propose a **Generalized Block-low-rank** (GBLR) matrix format, which includes many important structures such as Low-Rank (LR), Block Sparse (BSP), and Block-low-rank (BLR) matrices under some practical conditions. The new structured matrix format consists of two types of parameters: one guiding the *structure* of the matrix and the other specifying the *content* or the values of matrix elements. We show that the LR, BSP and BLR formats are special cases of the GBLR matrix. We also show that the GBLR format is closed under the interpolation between existing GBLR matrices in the structural parameter space, which we believe is a strong evidence that the GBLR format is able to capture undiscovered structured matrix formats.

2) We introduce a *differentiable* parameterization of the structural parameters – widths and locations of blocks – of the GBLR format. The structural parameters are defined in the *frequency* domain, and are processed by the proposed **Gaussian-Dirichlet** (Gaudi) function followed by inverse Fast Fourier Transform (IFFT) to the format named *Gaudi-GBLR*. We show that the derivatives of the Gaudi function with respect to the structural parameters exist almost everywhere, even when the width is zero.

3) We propose a practical learning algorithm based on the proximal gradient descent to train compact neural networks with Gaudi-GBLR matrices. The proposed method is extensively evaluated on the Vision Transformers (ViT) (Dosovitskiy et al., 2020) and MLP-Mixer (Tolstikhin et al., 2021), outperforming prior approaches using hand-designed structured matrices.

## 2 PRELIMINARIES

**Notation.** We use $\odot$ to indicate the elementwise (Hadamard) product of two matrices. The imaginary unit is denoted by $\imath = \sqrt{-1}$. The normalized sinc function is defined by $\mathrm{sinc}(x) = \frac{\sin \pi x}{\pi x}$ where $\mathrm{sinc}(0) := 1$. The (element-wise) floor function is denoted by $\lfloor \cdot \rfloor$. The index of elements in a matrix or vector starts from zero (instead of one), following the convention of Cooley & Tukey (1965). Also, $\mathcal{I}_n = \{0, 1, \ldots, n\}$ denotes the index set from 0 to $n$.

**Assumption.** For simplicity, we assume the weights are square matrices. Extension to rectangular matrix cases is discussed in Appendix A.3.

### 2.1 BLOCK-RELATED MATRICES.

A **block** $\boldsymbol{B} \in \mathbb{R}^{|R| \times |C|}$ of a matrix $\boldsymbol{W} \in \mathbb{R}^{n \times n}$ is a submatrix of $\boldsymbol{W}$ with *consecutive cyclic* row and column indices. For example, $R$ can be $\{n-2, n-1, 0, 1\}$ if $|R| = 4$. Also, we say two blocks $\boldsymbol{B}_1$ and $\boldsymbol{B}_2$ **overlap** if they have shared elements of $\boldsymbol{W}$.

Two well-known block-structured matrices are a *block sparse* (BSP) matrix and a *block-low-rank* (BLR)(Amestoy et al., 2015) matrix. Informally speaking, a matrix is block-sparse when non-zero elements are gathered in blocks and such blocks are sparse in the matrix. A block-low-rank matrix

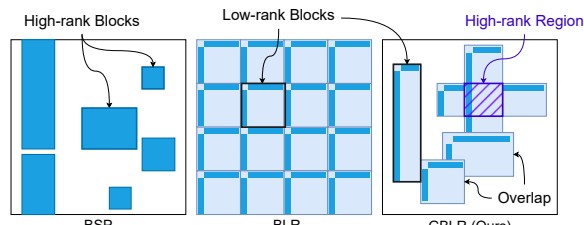

Figure 1: Comparison of block-sparse, block-low-rank, and our proposed Generalized block-low-rank matrices.

is composed of non-overlapping equally-partitioned low-rank blocks. Figure 1 illustrates the BSP and BLR matrices as well as our proposed generalized block-low-rank matrix, which we introduce in Section 3.1. We present the formal definitions of the block-sparse and block-low-rank matrices in Appendix A.1.

### 2.2 STRUCTURED MATRIX

We say a matrix $\boldsymbol{W} \in \mathbb{R}^{n \times n}$ is **structured** if, for any $\boldsymbol{x} \in \mathbb{R}^n$, the matrix-vector product (MVP) $\boldsymbol{y} = \boldsymbol{W}\boldsymbol{x}$ requires significantly less number of multiplications than $n^2$. For instance, LR, BSP, and BLR matrices are structured because the number of multiplications for MVP is determined by their (low) rank or sparsity. Although a general *sparse* matrix reduces the complexity of MVP to be a sub-quadratic function of $n$, it is excluded in our discussion of structured matrices because

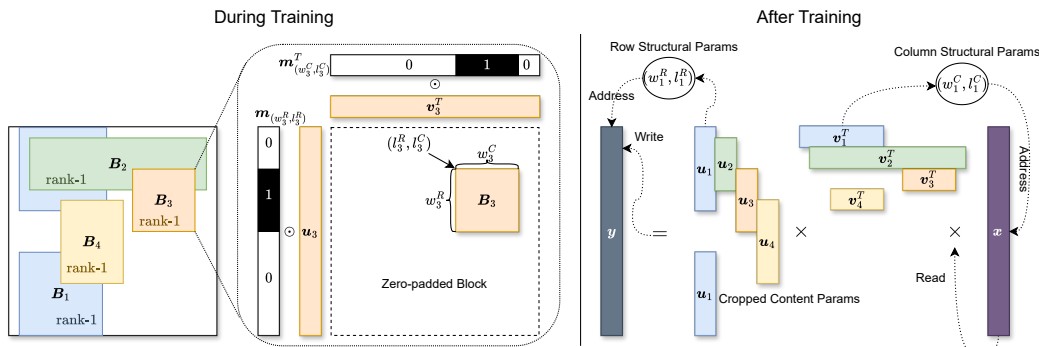

Figure 2: Left: An example of a GBLR matrix with 4 blocks. A block is generated from the *structural* parameters $(w^R, l^R), (w^C, l^C)$ and the *content* parameters $(\boldsymbol{u}, \boldsymbol{v})$, where $(w^R, l^R)$ and $(w^C, l^C)$ form binary masks $\boldsymbol{m}_{(w^R, l^R)}$ and $\boldsymbol{m}_{(w^C, l^C)}$, respectively. Note that overlapped regions can have a rank higher than one. Right: Efficient Matrix-Vector Product computations using cropped content parameters and structural parameters. The structural parameters locate the input and output indices/addresses to read and write.

processing moderately sparse matrices (10∼50% density) in conventional hardware such as GPUs does not reduce the actual run-time due to its unstructured positions of non-zero elements (Chen et al., 2022).

## 3 PROPOSED METHOD

We are interested in training a deep neural network (DNN) $f$ under a *computational cost* constraint:

$$\min_f \sum_{\boldsymbol{x}, \boldsymbol{y} \sim \mathcal{D}} \mathcal{L}(f(\boldsymbol{x}), \boldsymbol{y}) \quad \text{s.t.} \quad \text{cost}(f) \leq B, \tag{1}$$

where the first term is the cross-entropy loss for a classification task at a data point $\boldsymbol{x}$ and a label $\boldsymbol{y}$, and the constraint $\text{cost}(f)$ is the number of multiplications to compute the neural network output $f(\boldsymbol{x})$. Our method learns weight matrices of DNNs in a *generalized* structured matrix format in a *differentiable* manner.

### 3.1 GENERALIZED BLOCK-LOW-RANK (GBLR) MATRIX

We introduce a concept of a generalized structured matrix format that explains multiple popular structure matrices used in DNN training. The idea is that a block in a matrix can be expressed by a **sum of rank-1 blocks**, i.e., a rank-$r$ block is a sum of $r$ rank-1 blocks at the same position. In this manner, LR, BSP, and BLR matrices can be expressed under a unified framework (Theorem 1).

To be specific, our proposed structured matrix format is a generalized version of Block-Low-Rank (BLR) matrices. An $n$-by-$n$ *Generalized Block-Low-Rank* (GBLR) matrix $\boldsymbol{W}$ is obtained by *overlapping* multiple rank-1 blocks of *different* sizes at *arbitrary* locations, as depicted in Figure 2 (Left). The locations of the blocks as well as their element values are learned *simultaneously* from training data without explicit size or location restrictions.

Suppose there are $K$ blocks: $\boldsymbol{B}_1 \in \mathbb{R}^{w_1^R \times w_1^C}, \boldsymbol{B}_2 \in \mathbb{R}^{w_2^R \times w_2^C}, \ldots, \boldsymbol{B}_K \in \mathbb{R}^{w_K^R \times w_K^C}$. Each block $\boldsymbol{B}_k$ has two parameter sets: 1) the *structural* parameters that identify the *position* of the block in the row and column index set $\mathcal{I}_{n-1} = \{0, 1, \ldots, n-1\}$, and 2) the *content* parameters which specify the actual values of matrix elements.

**Configuration of structural parameters.** The position of a block is given by the indices of the rows and columns it occupies in the $n \times n$ matrix. Hence, the placement of a rectangle block can be identified by four numbers: *width* and *location* in terms of the row or column indices. Hence, we use a *location* parameter $l$ and a *width* parameter $w$ as the structural parameters of a block.

Figure 2 (Left) illustrates a block of size $w^R \times w^C$ in an $n \times n$ matrix at location $(l^R, l^C)$. The row (column) index set of a block is the sequence of numbers from $l^R$ ($l^C$) to $l^R + w^R$ ($l^C + w^C$) where the addition is a cyclic/modulo addition. For each block $\boldsymbol{B}_k$ for $k = 1, \ldots, K$, we have four parameters: $(w_k^R, l_k^R)$ for the row and $(w_k^C, l_k^C)$ for the column. We use the notation $\phi_k^R = (w_k^R, l_k^R)$ and $\phi_k^C = (w_k^C, l_k^C)$ to represent the tuple of width and location for the row ($\phi_k^R$) and column ($\phi_k^C$).

Based on the structural parameter $w_k^C$ and $l_k^C$, one can construct an $n$-dimensional binary mask that has $w_k^C \in \mathcal{I}_n$ consecutive ones starting from $l_k^C \in \mathcal{I}_{n-1}$ in the cyclic order:

$$m_{\phi_k^C}[j] = m_{(w_k^C, l_k^C)}[j] = \begin{cases} 1 & \text{if } l_k^C \leq j + an < l_k^C + w_k^C \\ 0 & \text{otherwise} \end{cases}, \; j \in \mathcal{I}_{n-1}, \; a \in \{0, 1\}, \quad (2)$$

where $a \in \{0, 1\}$ is necessary to make the order cyclic. We call the mask in Eq. 2 the *boxcar* mask since the non-zero elements are located consecutively. The boxcar mask is used to select $w_k^C$ (cyclic) consecutive non-zero elements of an $n$-dimensional vector. The mask for the rows $\boldsymbol{m}_{\phi_k^R}$ is obtained in the same way from $w_k^R$ and $l_k^R$.

**Configuration of content parameters.** To represent the values of a rank-1 block $\boldsymbol{B}_k$, we use two $n$-dimensional vectors $\boldsymbol{u}_k$ and $\boldsymbol{v}_k$ as *content* parameters along with the boxcar masks $\boldsymbol{m}_{\phi_k^R}, \boldsymbol{m}_{\phi_k^C}$. All these parameters $(\phi_k^R, \phi_k^C, \boldsymbol{u}_k, \boldsymbol{v}_k)$ are learned during the DNN training *simultaneously*. Since the boxcar masks $\boldsymbol{m}_{\phi_k^R}$ and $\boldsymbol{m}_{\phi_k^C}$ guide the location of the block in the $n \times n$ matrix, $\boldsymbol{u}_k$ and $\boldsymbol{v}_k$ are element-wise multiplied with the boxcar masks:

$$\text{ZeroPad}(\boldsymbol{B}_k) = (\boldsymbol{m}_{\phi_k^R} \odot \boldsymbol{u}_k)(\boldsymbol{m}_{\phi_k^C} \odot \boldsymbol{v}_k)^T,$$

where the resulting $n \times n$ matrix is a zero-padded block. Ideally, we expect the mask to expand / shrink and shift to find the right subset of the elements of a content parameter, while the content parameter updates the value of the elements selected by the mask.

Now we formally define the **Generalized Block-low-rank** (GBLR) format, which is the sum of $K$ zero-padded blocks:

$$\boldsymbol{W} = \sum_{k=1}^{K} (\boldsymbol{m}_{\phi_k^R} \boldsymbol{m}_{\phi_k^C}^T) \odot (\boldsymbol{u}_k \boldsymbol{v}_k^T) = \sum_{k=1}^{K} \left( \boldsymbol{m}_{\phi_k^R} \odot \boldsymbol{u}_k \right) \left( \boldsymbol{m}_{\phi_k^C} \odot \boldsymbol{v}_k \right)^T. \quad (3)$$

A GBLR matrix is associated with an average width $\bar{w} = \frac{1}{2K} \sum_{k=1}^{K} w_k^R + w_k^C$.

**Definition 1.** *Let* $\text{GBLR}(n, K, s)$ *be the set of matrices obtained by Eq. 3 for the average width less than or equal to* $s \geq 0$, *i.e.,* $\bar{w} = \frac{1}{2K} \sum_{k=1}^{K} w_k^R + w_k^C \leq s$. *A matrix* $\boldsymbol{W}$ *is an* $(n, K, s)$-*GBLR if* $\boldsymbol{W} \in \text{GBLR}(n, K, s)$.

We use the notation $\boldsymbol{\phi}(\boldsymbol{W}) := (\boldsymbol{w}(\boldsymbol{W}), \boldsymbol{l}(\boldsymbol{W}))$ to indicate the collection of the structural parameters of $\boldsymbol{W}$, where $\boldsymbol{w}(\boldsymbol{W}) = \{w_1^R, w_1^C, w_2^R, w_2^C, \ldots, w_K^R, w_K^C\}$ and $\boldsymbol{l}(\boldsymbol{W}) = \{l_1^R, l_1^C, l_2^R, l_2^C, \ldots, l_K^R, l_K^C\}$. We simply use $\boldsymbol{\phi} := \boldsymbol{\phi}(\boldsymbol{W})$ if $\boldsymbol{W}$ is clearly inferred in the context.

**Efficiency.** Once the structural parameters are fixed, it is unwise to store and use two $n$-dimensional content parameters for each block $\boldsymbol{B}_k$ because only $w_k^R$ elements of $\boldsymbol{u}_k$ and $w_k^C$ elements of $\boldsymbol{v}_k$ are non-zero according to the boxcar masks in Eq. 3. Hence, one can store and use the *cropped* content parameters $\boldsymbol{u}_{l^R:l^R+w^R}$ and $\boldsymbol{v}_{l^C:l^C+w^C}$ for MVP between an input $\boldsymbol{x} \in \mathbb{R}^n$ (which can be also cropped from $l^C$ to $l^C + w^C$) and a block $\boldsymbol{B}$, as described below and in Figure 2 (Right):

$$\text{ZeroPad}(\boldsymbol{B})\boldsymbol{x} = (\boldsymbol{m}_{(w^R, l^R)} \odot \boldsymbol{u})(\boldsymbol{m}_{(w^C, l^C)} \odot \boldsymbol{v})^T \boldsymbol{x}$$
$$= \text{ZeroPad}\left( \boldsymbol{u}_{l^R:l^R+w^R}(\boldsymbol{v}_{l^C:l^C+w^C}^T \boldsymbol{x}_{l^C:l^C+w^C}) \right),$$

which requires only $w^R + w^C$ multiplications. Hence, the number of multiplications (denoted by FLOPs) for multiplying $\boldsymbol{W} \in \text{GBLR}(n, K, s)$ with $\boldsymbol{x} \in \mathbb{R}^n$ is bounded by $2Ks$:

$$\text{FLOPs} = \sum_{k=1}^{K} (w_k^R + w_k^C) = 2K\bar{w} \leq 2Ks. \quad (4)$$

**Expressiveness.** Low-rank (LR), block sparse (BSP), and block-low-rank (BLR) matrices are popular structured matrices in the DNN literature, and they are special cases of the GLBR matrix under mild conditions. Proofs and formal definitions of LR, BSP, and BLR matrices are in Appendix A.1

**Theorem 1.** *Let $n, K, s$ be positive integers satisfying $Ks \geq n$. Then any $n$-by-$n$ rank-$\frac{Ks}{n}$ matrices and $(n, \frac{K}{s}, s)$-block-sparse matrices are $(n, K, s)$-GBLR. Also, any $(n, K, s, 1)$-block-low-rank matrices are $(n, K, s)$-GBLR if $K = (n/s)^2$.*

More importantly, a new structured matrix obtained by interpolating the structural parameters of two $(n, K, s)$-GBLR matrices is still $(n, K, s)$-GBLR, based on Theorem 2. Therefore, a new type of structured matrices can be derived from a set of GBLR matrices.

**Theorem 2** (Closed under structural interpolation). *Given two $n \times n$ matrices $\boldsymbol{W}, \boldsymbol{Z} \in$ GBLR$(n, K, s)$, and $\alpha \in [0, 1]$, consider the following combination between the structural parameters:*

$$\boldsymbol{w}' = \lfloor \alpha \boldsymbol{w}(\boldsymbol{W}) + (1 - \alpha)\boldsymbol{w}(\boldsymbol{Z}) \rfloor, \quad \boldsymbol{l}' = \lfloor \alpha \boldsymbol{l}(\boldsymbol{W}) + (1 - \alpha)\boldsymbol{l}(\boldsymbol{Z}) \rfloor.$$

*A matrix $\boldsymbol{Y}$ generated by Eq. 3 with the structural parameter $(\boldsymbol{w}', \boldsymbol{l}')$ is a $(n, K, s)$-GBLR matrix, $\boldsymbol{Y} \in$ GBLR$(n, K, s)$.*

Theorem 1 and Theorem 2 tell us that $(n, K, s)$-GBLR matrices cover a wide range of popular existing structured matrices and also undiscovered ones. In the following section, we introduce a differentiable tool to find/learn structured matrices in GBLR$(n, K, s)$.

## 3.2 GAUSSIAN-DIRICHLET (GAUDI) MASK

The matrix structure in the GBLR format is determined by the width and location parameters. We aim to extract/learn these structural parameters from training data using stochastic gradient descent. In order to do so, the non-differentiability of the boxcar mask parameters needs to be handled.

To tackle this issue, we introduce a *Dirichlet-kernel-based* parameterization of the boxcar to *explicitly* parameterize the width $w$ and location $l$ in the expression. Consider a boxcar mask of length $n$, width $w$, and location $l$. Let $\hat{m}_{(w,l)}$ be the discrete Fourier transform (DFT) of the mask $\boldsymbol{m}_{(w,l)}$. The frequency-domain representation of the boxcar mask is given by

$$\hat{m}_{(w,l)}[k] = e^{-2\pi i \frac{k}{n} l} \hat{m}_{(w,0)}[k] = e^{-2\pi i \frac{k}{n} l} \sum_{j=0}^{w-1} e^{-2\pi i j \frac{k}{n}} = e^{-2\pi i \frac{k}{n} l} w \frac{\text{sinc}\left(w \frac{k}{n}\right)}{\text{sinc}\left(\frac{k}{n}\right)} e^{i\pi k\left(\frac{1-w}{n}\right)} \quad (5)$$

$$= e^{-2\pi i \frac{k}{n} l} d_w[k],$$

where $d_w[k] := w \frac{\text{sinc}\left(w\frac{k}{n}\right)}{\text{sinc}\left(\frac{k}{n}\right)} e^{i\pi k\left(\frac{1-w}{n}\right)}$ is the *Dirichlet kernel of order $n$* (Bruckner et al., 1997) in the discrete domain $\mathcal{I}_{n-1} = \{0, 1, \ldots, n-1\}$.

Furthermore, we propose a smooth version of the Dirichlet-kernel-based mask by convolving the time-domain boxcar mask with the Gaussian-shape function. It is obtained in the frequency domain by element-wise multiplying the Gaussian function $g_\sigma[k] = \exp\left(-\frac{k^2}{2\sigma^2}\right)$ with the standard deviation $\sigma > 0$ to the Dirichlet kernel. We call the resulting function the **Gaussian-Dirichlet** (Gaudi) function $\mathbf{d}_w^\sigma$:

$$\mathbf{d}_w^\sigma[k] := g_\sigma[k] \cdot \mathbf{d}_w[k],$$

$$\tilde{\boldsymbol{m}}_{(w,l)}^\sigma := \textbf{IDFT}(\mathbf{d}_w^\sigma \cdot e^{-2\pi i \frac{\boldsymbol{k}}{n} l}), \quad (6)$$

where $\boldsymbol{k} = [0, 1, \ldots, n-1]$. And we call the mask generated by a Gaudi function $\tilde{\boldsymbol{m}}_{(w,l)}^\sigma$ the **Gaudi mask**, where the parameter $\sigma$ controls the smoothness. Note that as $\sigma \to \infty$, the Gaudi mask converges to the boxcar mask (Lemma 5). Figure 3 visualizes the time-domain Gaudi mask approaching the boxcar mask.

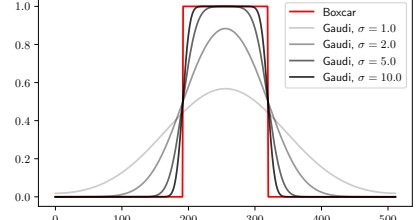

Figure 3: Comparison between Boxcar mask and Gaudi masks in the time domain with different smoothing factors $\sigma$. The Gaudi mask converges to the Boxcar mask as $\sigma$ grows.

A useful property of the Gaudi mask is that one can obtain exact derivatives with respect to the width parameter, even when the width is zero. To show it, we relax the domain of widths and locations to the continuous interval $[0, n]$ (see Appendix A.4).

**Theorem 3.** *Let $n < \infty$ be a finite positive integer. For any $\sigma \in (0, \infty]$ and $w, l \in [0, n]$, the derivatives of the Gaudi mask $\tilde{\boldsymbol{m}}^{\sigma}_{(w,l)}$ with respect to $w$ and $l$ are bounded almost everywhere:*

$$\left\| \frac{\partial \tilde{\boldsymbol{m}}^{\sigma}_{(w,l)}}{\partial w} \right\|_2 < \infty, \quad \left\| \frac{\partial \tilde{\boldsymbol{m}}^{\sigma}_{(w,l)}}{\partial l} \right\|_2 < \infty.$$

Especially when $w = 0$, the derivative with respect to $w$ is neither divergent nor zero.

**Corollary 4.** *For any $l \in [0, n]$ and $\sigma \in (0, \infty]$, the norm of the derivative of the Gaudi mask $\tilde{\boldsymbol{m}}^{\sigma}_{(w,l)}$ with respect to $w$ at $w = 0$ is well-defined and greater than zero, i.e., $0 < \left\| \frac{\partial \tilde{\boldsymbol{m}}^{\sigma}_{(w,l)}}{\partial w} \right\|_2 < \infty$.*

To allow learning the mask structural parameters in a differentiable manner, we plug the Gaudi mask (Eq. 6) into Eq. 3 as the mask of GBLR matrices to model **Gaussian-Dirichlet GBLR** (Gaudi-GBLR) with parameters $\boldsymbol{\theta} = (\boldsymbol{\phi}, \boldsymbol{U}, \boldsymbol{V}, \sigma)$:

$$\boldsymbol{W}^{\boldsymbol{\theta}} = \boldsymbol{W}^{(\boldsymbol{\phi}, \boldsymbol{U}, \boldsymbol{V}, \sigma)} = \sum_{k=1}^{K} \left( \tilde{\boldsymbol{m}}^{\sigma}_{\phi_k^R} \odot \boldsymbol{u}_k \right) \left( \tilde{\boldsymbol{m}}^{\sigma}_{\phi_k^C} \odot \boldsymbol{v}_k \right)^T. \tag{7}$$

In practice, one can use small $\sigma \approx 1$ at the beginning of the training process to update the corresponding (unmasked) content parameters which are more than necessary, then gradually increase $\sigma$ to adjust the content parameters with a tighter mask. Since the purpose of using a Gaudi mask is to learn structural parameters of the GBLR matrix by *Gradient Descent*, Gaudi-GBLR matrices are later replaced by GBLR matrices once the structural parameters are found/learned. To compute MVP using GBLR matrices, one can use the cropped content parameters and inputs, as we discussed in Section 3.1-efficiency, without constructing masks at all. Hence, during the inference, there is no overhead to compute Gaudi masks.

## 3.3 Learning Gaudi-GBLR for Efficient Neural Networks

We now introduce an algorithm to learn the structural parameters of Gaudi-GBLR matrices. The goal of the learning algorithm is to identify Gaudi-GBLR matrix structures (i.e., their parameters) that allow computationally efficient DNNs. Our discussion is centered around a two-layered multi-layer perceptron (MLP) for ease of understanding. However, the technique can be applied to general DNNs that incorporate linear layers.

Now let us consider a two-layered MLP $f^{\boldsymbol{\theta}}$ with a Gaudi-GBLR weight matrix $\boldsymbol{W}^{\boldsymbol{\theta}}$: $f^{\boldsymbol{\theta}}(\boldsymbol{x}) = \boldsymbol{h}^T a(\boldsymbol{W}^{\boldsymbol{\theta}} \boldsymbol{x} + \boldsymbol{b})$. We initially relax the domain of $w \in \mathcal{I}_n$ and $l \in \mathcal{I}_{n-1}$ of the Gaudi mask to the real-valued space $w, l \in [0, n]$ as we discuss in Appendix A.4. Due to the property of

---

**Algorithm 1** GBLR Learning by PGD

1: **repeat**
2:     $(\boldsymbol{x}_1, \boldsymbol{y}_1), \ldots, (\boldsymbol{x}_M, \boldsymbol{y}_M) \sim \mathcal{D}$
3:     $\ell \leftarrow \frac{1}{M} \sum_{i=1}^{M} \mathcal{L}(f^{\boldsymbol{\theta}}(\boldsymbol{x}_i), \boldsymbol{y}_i)$
4:     **for** trainable $p \in \boldsymbol{\theta}$ **do**
5:         $p \leftarrow p - \eta \cdot \text{AdamW}(\nabla \ell)$
6:     **end for**
7:     **for** $k = 1, 2, \ldots, K$ **do**
8:         $w_k \leftarrow \text{clip}_{0,n}(S_{\eta\lambda}(w_k))$
9:     **end for**
10: **until** converge

---

Gaudi-GBLR matrices Eq. 4, the computational cost constraint on the DNN $f^{\boldsymbol{\theta}}$ in Problem (1) can be replaced by a constraint on the sum of the width parameters of $\boldsymbol{W}^{\boldsymbol{\theta}}$. Specifically, we find the width parameters $\boldsymbol{w} = \{w_1^R, w_1^C, \ldots, w_K^R, w_K^C\}$ of $\boldsymbol{W}^{\boldsymbol{\theta}}$ satisfying $\|\boldsymbol{w}\|_1 \leq B$ since $\sum_{k=1}^{K} w_k^R + w_k^C = \|\boldsymbol{w}\|_1$. To solve the problem with Gradient Descent, we relax this $\ell_1$-norm constrained problem to a *unconstrained* one using the method of Lagrange multiplier:

$$\min_{\boldsymbol{\theta}} \sum_{(\boldsymbol{x}, \boldsymbol{y}) \sim \mathcal{D}} \mathcal{L}(f^{\boldsymbol{\theta}}(\boldsymbol{x}), \boldsymbol{y}) + \lambda \|\boldsymbol{w}\|_1, \quad \lambda \geq 0. \tag{8}$$

The resulting computational budget is implicitly constrained by a hyperparameter $\lambda \geq 0$.

Theorem 3 guarantees the derivatives of the widths and locations of the Gaudi-GBLR matrix in $f^{\boldsymbol{\theta}}$ can be obtained with any positive smoothing parameter $\sigma > 0$ so that we can safely learn the parameters in the continuous domain $[0, n]$. Specifically, we update the width parameter $\boldsymbol{w}$ in the $\ell_1$-norm term in Problem (8) by Proximal Gradient Descent (PGD):

$$\boldsymbol{w}_{t+1} = \text{clip}_{0,n}(S_{\eta\lambda}(\boldsymbol{w}_t - \eta \nabla \mathcal{L}(f^{\boldsymbol{\theta}}(\boldsymbol{x}), \boldsymbol{y}))), \tag{9}$$

where $S_\mu(x) = \begin{cases} \text{sign}(x) \cdot (|x| - \mu) & \text{if } x > \mu \\ 0 & \text{otherwise} \end{cases}$ is the element-wise soft shrinkage function and $\text{clip}_{0,n}(\cdot)$ clamps the elements of the inputs to $[0, n]$. In practice, the gradient is calculated with an adaptive optimizer such as AdamW (Loshchilov & Hutter, 2017) (see Line 5 in Algorithm 1). The overall process is summarized in Algorithm 1. Although Problem (8) is non-linear, our experimental results show that PGD can attain good local optima with an adaptive optimizer and the initialization method we propose in Appendix A.2. A practical learning method for the width and location parameters defined in the discrete spaces $\mathcal{I}_n$ and $\mathcal{I}_{n-1}$ is discussed in Appendix A.5.

## 4 EXPERIMENTS

We evaluate our proposed method by replacing the weight matrices of Vision Transformers (ViT) (Dosovitskiy et al., 2020), and MLP-Mixer (Tolstikhin et al., 2021) with Gaudi-GBLR matrices. For the experiment, we set the number of blocks $K$ equal to the number of columns of the matrix $n$. We also evaluate alternative schemes for comparisons where the weights are replaced by popular hand-designed structured matrix formats such as Low-Rank (LR), Block-Sparse-plus-Low-Rank (BSP-LR), and Block-low-rank (BLR). For LR matrices, we use singular vector decomposition to find the best rank-$s$ approximation of the given matrix for the predefined rank of $s$. Pixelfly (Chen et al., 2022) and Monarch (Dao et al., 2022) are schemes that use BSP-LR and BLR matrices respectively. We set the structural parameters of these alternative schemes to exhibit similar computational costs for MVP compared to our proposed scheme. Note that the structural parameter sets for LR, BSP-LR, and BLR do not change across different layers in the neural network. We denote the number of multiplications by FLOPs, and use 8 NVIDIA A40 GPUs in our experiments. The detailed experimental settings are described in Appendix A.7. Our source code is available at `https://github.com/changwoolee/Gaudi-GBLR`.

### 4.1 FINE-TUNING RESULTS

**Vision Task.** We use the *pre-trained* weights of the ViT-Base on ImageNet and initialize the parameters of the Gaudi-GBLR matrices by Algorithm 2 in Section A.2. The ViTs with Gaudi-GBLR matrices were fine-tuned on ImageNet (Russakovsky et al., 2015). During the initialization, we set the computational budget of all matrices in the network the same. For a fair comparison, the same set of hyperparameters was used throughout our fine-tuning experiments.

The highest accuracy is achieved by Gaudi-GBLR in ViT-Base with a patch size of $16 \times 16$ on the ImageNet validation dataset after fine-tuning it for 35 epochs. Figure 4 shows that Gaudi-GBLR preserves the accuracy well when the complexity is reduced to 30% of the original 'Dense' model which does not use structured matrices (its FLOPs count is normalized to 1).

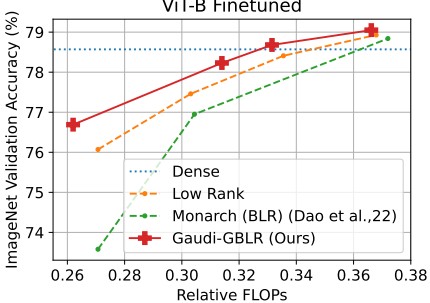

Figure 4: ImageNet accuracy after fine-tuning ViT-Base weights replaced with structured weight matrices. Dense: the original ViT-Base model.

The other hand-designed approaches exhibit more significant accuracy degradations for the same complexity reduction. Overall, Gaudi-GBLR strikes better Accuracy-FLOPs trade-offs than LR or Monarch approaches. The higher accuracy for a similar FLOPs count quantifies the gain from the learned structured matrices. In addition to ViT, we also tested GBLR matrices on ResNet (He et al., 2016). Readers interested in further details can find the results in Appendix A.8.

**Language Task.** We test the GBLR matrix to the weights of the pre-trained GPT-2 (Radford et al., 2019). We compare our method to LR and BLR (Monarch) matrices. We evaluated the perplexity of WikiText103 (Merity et al., 2016) validation set after 10 epochs of fine-tuning on the training set. The perplexity of each model is in Table 1. Our method

Table 1: Perplexity by weight type of GPT-2 after fine-tuning on WikiText103.

| Weight Type | Perplexity ($\downarrow$) | Relative FLOPs |
|---|---|---|
| Dense | 19.36 | 100% |
| Low Rank | 19.48 | 43.75% |
| Monarch | 20.56 | 43.75% |
| Gaudi-GBLR | **19.24** | **43.7**% |

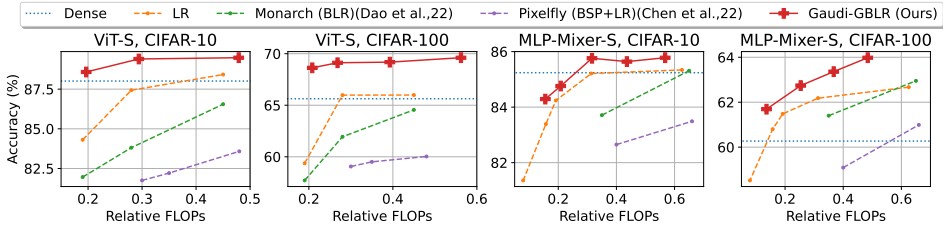

Figure 5: Accuracy-Cost trade-off of models trained from scratch on CIFAR-10/100 dataset.

Table 2: ImageNet accuracies and GFLOPs on a 224 × 224 image of ViT-Base models trained from scratch with dense matrices and Gaudi-GBLR matrices.

| Model | Acc. (%) | GFLOPs |
|---|---|---|
| ViT-Base | 78.57 | 17.2 |
| w/ Gaudi-GBLR | 78.51 | 5.65 |

Table 3: FLOPs by type of linear layers with Gaudi-GBLR matrix of ViT-Base trained on ImageNet. Units in $10^3$ FLOPs.

| Layer Type | Min | Max | Avg |
|---|---|---|---|
| Query | 4.2 | 67.6 | 29.0 |
| Key | 11.6 | 87.6 | 42.1 |
| Value | 34.2 | 135.5 | 95.0 |
| MLP-FC1 | 359.8 | 910.2 | 597.7 |
| MLP-FC2 | 349.4 | 1,175.2 | 756.6 |

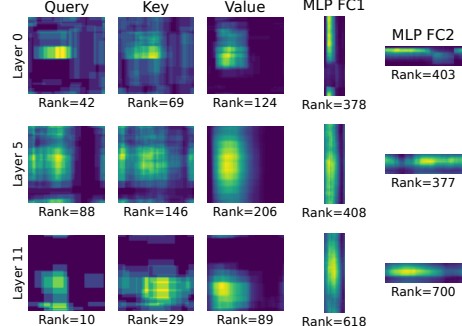

Figure 6: Mask patterns of weights for Query, Key, Value, FC1 and FC2 in Layer 0, 5, and 11 (out of 0 ∼ 11). The brighter, the more overlapped blocks. The numbers below indicate the rank of the matrix (max: 768).

achieved the lowest perplexity, outperforming the dense baseline and utilizing the smallest number of FLOPs.

## 4.2 TRAINING FROM SCRATCH

We train ViTs (Dosovitskiy et al., 2020) and MLP-Mixers (Tolstikhin et al., 2021) with structured weight matrices on CIFAR-10&100 (Krizhevsky et al., 2009) by Algorithm 1 from randomly-initialized content parameters. We set $\sigma = 1$ for the first epoch and gradually increase it to 100 until the training process ends (see Appendix A.7). In Figure 5, we study the accuracy-FLOPs trade-off using CIFAR10/100 datasets when the models are trained from scratch (i.e., not fine-tuned from pre-trained weights). As in the ImageNet fine-tuning experiment, Gaudi-GBLR achieves superior accuracy-FLOPs trade-offs outperforming the other hand-designed structured matrices.

## 4.3 ANALYSIS ON LEARNED GAUDI-GBLR MATRICES

In this section, we study the learned Gaudi-GBLR matrices in terms of computational budget allocation, and mask patterns. The analysis is based on ViT-Base Gaudi-GBLR matrices trained on ImageNet from scratch by following the same $\sigma$ adaptation rule used in CIFAR10/100 experiments. The accuracy and FLOPs of the ViT-Base with Gaudi-GBLR is reported in Table 2.

**Learned Computational Budget Allocation.** The proposed learning framework automatically allocates the computational budget to all linear layers of ViT-Base during the training process given by Algorithm 1. The algorithm finds a well-balanced (not necessarily equal) allocation for each matrix to meet the overall cost budget while minimizing the loss function. As a result, Gaudi-GBLR weight matrices in the network have unequal widths requiring different FLOPs for each MVP. Table 3 summarizes the min/max/average FLOPs statistics collected for different types of linear layers. Within an Attention layer, the weights for *Values* have the highest budget whereas *Queries* and *Keys* use a smaller budget. The smallest layer uses only about 4,200 FLOPs for MVP involving a

$768 \times 768$ matrix and a $768 \times 1$ vector. The FLOPs assigned to the linear layer of channel MLPs (*MLP-FC1* and *MLP-FC2* in Table 3) vary significantly. Although the size of the weight matrix used in the MLPs are $4\times$ larger than the ones used for Value, the *MLP-FC2*-type layers use $7.96\times$ FLOPs than the *Value*-type layers in average.

**Visualization.** Figure 6 visualizes the locations of the blocks in exemplary Gaudi-GBLR weight matrices of ViT-Base trained from scratch. We select the first, middle, and the last layers of different types: linear layers for Query, Key, Values in Attention modules, and two linear layers (FC1 and FC2) in MLPs. Bright colors in Figure 6 highlight regions where masks are overlapped. Interestingly, the resulting matrix is neither BSP nor BLR. It is observed that blocks are concentrated in a small number of regions. We believe this is related to the Multi-Head (Vaswani et al., 2017) scheme of the ViT. Each weight matrix of an attention layer is a collection of weights for multiple heads. Because some heads are more important than others, more blocks are allocated to those regions. Notice the rank of matrices obtained from the GBLR framework differs significantly across different layers and matrix types (Values, Queries, and Keys).

## 5 RELATED WORKS

**Structured Matrices for DNNs.** Prior works have focused on manually designing suitably structured matrices for DNNs since explaining the structure of every weight matrix in practical DNNs such as Transformers (Vaswani et al., 2017) is challenging. Hsu et al. (2022) used weighted low-rank decomposition for the weights of language models. Butterfly matrices (Li et al., 2015; Dao et al., 2019) were adopted in the form of Block-Sparse (BSP) format (Pixelfly) by Chen et al. (2022), and also in the form of Block-low-rank (BLR) format (Monarch) by Dao et al. (2022). In contrast, our method learns the structure of weight matrices from the training data. **Layer-wise Low-Rank DNN Compression** has been studied in terms of Augmented Lagrangian Multiplier (Idelbayev & Carreira-Perpinán, 2020), subspace grouping (Liebenwein et al., 2021), and quadratic programming (Li & Shi, 2018). However, they work only on the low-rank format, and involve multiple loops and SVD at every iteration, whereas ours can find layer-wise structure and budget in a single loop.

**Mask Learning.** For neuron pruning/selection, a binary mask with surrogate gradient has been adopted to learn its pattern. Jang et al. (2016) and Maddison et al. (2017) propose alternative distributions close to the Bernoulli distribution, which allow the continuous gradient. Movement Pruning (Sanh et al., 2020) utilizes Straight-Through Estimator (STE) (Hinton, 2012; Bengio et al., 2013; Zhou et al., 2016) to pass the gradient through the Top-$k$ function. Lin et al. (2017) adopts deep reinforcement learning to select the input-dependent mask. On the contrary, our mask design solves the non-existing gradient problem by Gaudi-function-based parameterization in the frequency domain.

**One-shot Neural Architecture Search.** Neural Architecture Search (NAS) (Zoph & Le, 2016; Liu et al., 2018) seeks the optimal neural network structures from training data. To improve search efficiency, Pham et al. (2018); Liu et al. (2018) adopt the one-shot NAS technique that selects sub-network candidates from a super-network. Our method also falls into a similar category of finding a small-sized neural network in the scope of a structured matrix format.

**Frequency-domain Learning.** DiffStride (Riad et al., 2022) learns a stride of the *pooling* operation for images by cropping a rectangular region of the frequency response of an image. They utilize an approximated boxcar mask for a differentiable stride. Although DiffStride shares similar components with Gaudi masks, the fundamental difference is in the design of the mask. We parameterize the mask in the *frequency* domain where widths and locations inherit the exact gradient.

## 6 CONCLUSION

We propose a generalized and differentiable framework for learning structured matrice for efficient neural networks by gradient descent. We introduce a new generalized format of structured matrices and parameterize the structure in the frequency domain by the Gaussian-Dirichlet (Gaudi) function with a well-defined gradient. Effective learning algorithms are provided for our framework showing flexibility and differentiability to find expressive and efficient structures from training data in an end-to-end manner. Evaluation results show that the proposed framework provides the most efficient and accurate neural network models compared to other hand-designed popular structured matrices.

## 7 REPRODUCIBILITY STATEMENT

The authors make the following efforts for reproducibility: 1) We release our code at `https://github.com/changwoolee/Gaudi-GBLR`, 2) we provide the detailed settings and hyperparameters in Section 4, A.5, and A.7, and 3) the proofs of all theorems, lemmas, and corollaries are presented in Section A.1.

## ACKNOWLEDGMENT

We thank Sara Shoouri, Pierre Abillama, Andrea Bejarano-Carbo, and Mingyu Yang for the insightful feedback on the paper. This work was supported in part by COGNISENSE, one of seven centers in JUMP 2.0, a Semiconductor Research Corporation (SRC) program sponsored by DARPA.

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

# A  APPENDIX

## A.1  PROOFS

In this section, we provide the missing definitions, propositions, and proofs. The original theorems are restated for completeness.

### A.1.1  DEFINITIONS

We introduce formal definitions of the block-related matrices discussed in our paper.

**Definition 2** (Block-sparse matrix). *An $n$-by-$n$ matrix is $(n, m, s)$-**block-sparse (BSP)** if it contains $m$ non-overlapping non-zero blocks whose dimension is at most $s \times s$.*

A $(n, m, s)$-block-sparse matrix may contain blocks of different sizes, but each of them has maximum $s \times s$ elements.

**Definition 3** (Block-low-rank matrix (Amestoy et al., 2015; Jeannerod et al., 2019)). *An $n$-by-$n$ matrix is $(n, m, s, r)$-**block-low-rank (BLR)** if, when it is equally partitioned into $m = p \times p$ non-overlapping blocks of dimension $s \times s$, every block has a rank at most $r \ll \frac{n}{p}$.*

The $(n, m, s, r)$-BLR matrix has blocks of the same size that tile the entire matrix.

### A.1.2  PROOF OF THEOREM 1

**Theorem 1.** *Let $n, K, s$ be positive integers satisfying $Ks \geq n$. Then any $n$-by-$n$ rank-$\frac{Ks}{n}$ matrices and $(n, \frac{K}{s}, s)$-block-sparse matrices are $(n, K, s)$-GBLR. Also, any $(n, K, s, 1)$-block-low-rank matrices are $(n, K, s)$-GBLR if $K = (n/s)^2$.*

*Proof.* It is sufficient to find the structural parameters of an $(n, K, s)$-GBLR matrix for each structured matrix format. Let $\boldsymbol{w}^R, \boldsymbol{w}^C \in \mathcal{I}_n^K$ be the width parameters of rows and columns of the $(n, K, s)$-GBLR matrix, respectively, and $\boldsymbol{l}^R, \boldsymbol{l}^C \in \mathcal{I}_{n-1}^K$ be the locations parameters of rows and columns, respectively, where $\mathcal{I}_n = \{0, 1, \ldots, n\}$.

**Low Rank Matrix.**  The rank-$\frac{Ks}{n}$ matrix is equivalent to the GBLR matrix with $Ks/n$ full-sized rank-1 blocks, i.e., $\boldsymbol{w}^R = \boldsymbol{w}^C = [\underbrace{n, n, \ldots, n}_{Ks/n}, 0, 0, \ldots, 0]$ and $\boldsymbol{l}^R = \boldsymbol{l}^C = \boldsymbol{0}$.

**Block Sparse Matrix.**  A block of the $(n, \frac{K}{s}, s)$-block-sparse matrix can have full rank. Consider a rank-1 block $\boldsymbol{B}$ of a $(n, K, s)$-GBLR matrix with some location parameters where the sum of the row width and the column width is $2s$. The maximum dimension of the block $\boldsymbol{B}$ is $s \times s$. Then clone and overlap the identical block with the same location and width parameters $s$ *times* to form a *full-rank* block of the dimension at most $s \times s$. Since the full-rank block can be generated $K/s$ times, any $(n, K/s, s)$-block-sparse matrix can be modeled by the corresponding width parameters and the location parameters.

**Block Low rank Matrix.**  Since $K = \frac{n^2}{s^2}$, let us divide the $n$-by-$n$ matrix into $(n/s)^2$ blocks of the dimension of $s$-by-$s$. Assign the width and the location parameters correspondingly. The resulting matrix forms the $(n, K, s, 1)$-block-low-rank matrix. $\qquad\square$

### A.1.3  PROOF OF THEOREM 2

**Theorem 2** (Closed under structural interpolation). *Given two $n \times n$ matrices $\boldsymbol{W}, \boldsymbol{Z} \in$ GBLR$(n, K, s)$, and $\alpha \in [0, 1]$, consider the following combination between the structural parameters:*

$$\boldsymbol{w}' = \lfloor \alpha \boldsymbol{w}(\boldsymbol{W}) + (1 - \alpha)\boldsymbol{w}(\boldsymbol{Z}) \rfloor, \quad \boldsymbol{l}' = \lfloor \alpha \boldsymbol{l}(\boldsymbol{W}) + (1 - \alpha)\boldsymbol{l}(\boldsymbol{Z}) \rfloor.$$

*A matrix $\boldsymbol{Y}$ generated by Eq. 3 with the structural parameter $(\boldsymbol{w}', \boldsymbol{l}')$ is a $(n, K, s)$-GBLR matrix, $\boldsymbol{Y} \in$ GBLR$(n, K, s)$.*

*Proof.* Let $w'^R_k$ and $w'^C_k$ be the width of the row and the column of the $k$th block of $\boldsymbol{Y}$, respectively. We use the same style of notation for $w(\boldsymbol{W})^R_k, w(\boldsymbol{W})^C_k$ and $w(\boldsymbol{Z})^R_k, w(\boldsymbol{Z})^C_k$ to denote the row and column width of the $k$th block of $\boldsymbol{W}$ and $\boldsymbol{Z}$. The interpolated matrix $\boldsymbol{Y}$ is $(n, K, s)$-GBLR if and only if the mean of the elements of $\boldsymbol{w}'$ is less than or equal to $s$, or equivalently the sum of the elements is less than or equal to $2Ks$. Also, both $\boldsymbol{W}$ and $\boldsymbol{Z}$ have $2Ks$ sum of the width parameters. From this, the sum of the interpolated width is less than or equal to $2Ks$:

$$
\begin{aligned}
\sum_i \boldsymbol{w}'_i &= \sum_{k=1}^K w'^R_k + w'^C_k \\
&= \sum_{k=1}^K \lfloor \alpha w(\boldsymbol{W})^R_k + (1-\alpha)w(\boldsymbol{Z})^R_k \rfloor + \lfloor \alpha w(\boldsymbol{W})^C_k + (1-\alpha)w(\boldsymbol{Z})^C_k \rfloor \\
&\leq \sum_{k=1}^K \alpha w(\boldsymbol{W})^R_k + (1-\alpha)w(\boldsymbol{Z})^R_k + \alpha w(\boldsymbol{W})^C_k + (1-\alpha)w(\boldsymbol{Z})^C_k \\
&= \alpha \sum_{k=1}^K (w(\boldsymbol{W})^R_k + w(\boldsymbol{W})^C_k) + (1-\alpha) \sum_{k=1}^K (w(\boldsymbol{W})^R_k + w(\boldsymbol{W})^C_k) \\
&\leq \alpha 2Ks + (1-\alpha)2Ks \\
&= 2Ks.
\end{aligned}
$$

Hence the interpolated matrix $\boldsymbol{Y} \in \mathsf{GBLR}(n, K, s)$. $\qquad\square$

### A.1.4 PROOF OF LEMMA 5

**Lemma 5.** *Let $n \in \mathbb{N}$ be a positive integer. Let $l \in \mathcal{I}_{n-1}$ and $w \in \mathcal{I}_n$. Consider the discrete boxcar mask $\boldsymbol{m}_{(w,l)}$ with the structural parameter $(w,l)$ as in Eq. 2 and the Gaudi mask $\tilde{\boldsymbol{m}}^\sigma_{(w,l)}$ with the same parameters and $\sigma > 0$ as in Eq. 6. Then $\tilde{\boldsymbol{m}}^\sigma_{(w,l)}$ converges to $\boldsymbol{m}_{(w,l)}$ as $\sigma \to \infty$, where the bound between two at $\sigma$ is given as follows:*

$$
\frac{\|\boldsymbol{m}_{(w,l)} - \tilde{\boldsymbol{m}}^\sigma_{(w,l)}\|_2}{\|\boldsymbol{m}_{(w,l)}\|_2} \leq 1 - e^{-\frac{(1-n^{-1})^2}{2\sigma^2}}.
$$

*Proof.* By Parseval's theorem,

$$
\|\boldsymbol{m}_{(w,l)} - \tilde{\boldsymbol{m}}^\sigma_{(w,l)}\|_2^2 = \frac{1}{n} \|\hat{\boldsymbol{m}}_{(w,l)} - \hat{\tilde{\boldsymbol{m}}}^\sigma_{(w,l)}\|_2^2 \tag{10}
$$

$$
= \frac{1}{n} \sum_{k=0}^{n-1} \left| \mathrm{d}_w[k] - \mathrm{d}_w[k] e^{-\frac{k^2}{2\sigma^2}} \right|^2
$$

$$
= \frac{1}{n} \sum_{k=0}^{n-1} |\mathrm{d}_w[k]|^2 \left(1 - e^{-\frac{k^2}{2\sigma^2}}\right)^2
$$

$$
\leq \frac{1}{n} \sum_{k=0}^{n-1} |\mathrm{d}_w[k]|^2 \left(1 - e^{-\frac{\left(\frac{n-1}{n}\right)^2}{2\sigma^2}}\right)^2
$$

$$
= \|\boldsymbol{m}_{(w,l)}\|_2^2 \left(1 - e^{-\frac{\left(1-\frac{1}{n}\right)^2}{2\sigma^2}}\right)^2, \tag{11}
$$

where Eq. 11 is also due to Parseval's theorem.

$\qquad\square$

### A.1.5 PROOF OF THEOREM 3

Let us first consider the derivative of the sinc function in eq. 5.

**Lemma 6.** *For any $n \neq 0$,* $\frac{\mathrm{dsinc}\frac{wk}{n}}{\mathrm{d}w} = \frac{\cos\frac{\pi wk}{n}}{w} - \frac{\mathrm{sinc}\frac{wk}{n}}{w}$.

*Proof.*

$$
\begin{aligned}
\frac{\mathrm{dsinc}\frac{wk}{n}}{\mathrm{d}w} &= \frac{\mathrm{d}}{\mathrm{d}w}\frac{\sin\frac{\pi wk}{n}}{\frac{\pi wk}{n}} \\
&= -\left(\frac{\pi wk}{n}\right)^{-2}\frac{\pi k}{n}\sin\frac{\pi wk}{n} + \left(\frac{\pi wk}{n}\right)^{-1}\cos\frac{\pi wk}{n}\cdot\frac{\pi k}{n} \\
&= \frac{\cos\frac{\pi wk}{n}}{w} - \frac{\sin\frac{\pi wk}{n}}{\left(\frac{\pi wk}{n}\right)^2} \\
&= \frac{\cos\frac{\pi wk}{n}}{w} - \frac{\mathrm{sinc}\frac{wk}{n}}{w}.
\end{aligned}
$$

$\square$

Note that the derivative in Lemma 6 at $w \to 0$ is zero since $\lim_{t\to 0}\mathrm{sinc}'(t) = 0$.

The following lemma is useful for proving Theorem 3.

**Lemma 7.** *For any $\sigma \in (0, \infty]$ and $w, l \in [0, n]$, the derivative of the Gaudi function $\boldsymbol{d}^{\sigma}_{(w,l)}$ with respect to $w$ is as follows:*

$$
\frac{\partial d^{\sigma}_w[k]}{\partial w} = e^{-2\pi \imath \frac{k}{n}w}\cdot e^{\pi \imath \frac{k}{n}}\cdot e^{-\frac{k^2}{2\sigma^2}}\cdot\frac{1}{\mathrm{sinc}\frac{k}{n}}.
$$

*Proof.* Let us merge the terms of the Gaudi function $\mathrm{d}^{\sigma}_w[k]$ that does not contain $w$ into $C_k$:

$$
\begin{aligned}
\mathrm{d}^{\sigma}_w[k] &= e^{-\frac{k^2}{2\sigma^2}}\cdot w\cdot\frac{\mathrm{sinc}\frac{wk}{n}}{\mathrm{sinc}\frac{k}{n}}e^{\frac{\imath\pi k(1-w)}{n}} \\
&= w\cdot\mathrm{sinc}\frac{wk}{n}\cdot e^{\frac{-\imath\pi kw}{n}}\cdot C_k
\end{aligned}
$$

where $C_k = e^{\pi\imath\frac{k}{n}}e^{-\frac{k^2}{2\sigma^2}}\frac{1}{\mathrm{sinc}\frac{k}{n}}$. From Lemma 6, the derivative of the Gaudi function with respect to $w$ is as follows:

$$
\begin{aligned}
\frac{\partial}{\partial w}\mathrm{d}^{\sigma}_w[k] &= C_k\cdot\frac{\partial}{\partial w}\left(w\cdot\mathrm{sinc}\frac{wk}{n}\cdot e^{\frac{-\imath\pi kw}{n}}\right) \\
&= C_k\cdot\left(\mathrm{sinc}\frac{wk}{n}e^{-\pi\imath\frac{k}{n}w} + \left(\frac{\cos\frac{\pi wk}{n} - \mathrm{sinc}\frac{wk}{n}}{w}\right)we^{-\pi\imath\frac{k}{n}w} + w\mathrm{sinc}\frac{wk}{n}\cdot\left(-\pi\imath\frac{k}{n}\right)e^{-\pi\imath\frac{k}{n}w}\right) \\
&= C_k\cdot(\cos\frac{\pi wk}{n} - \imath\sin\frac{\pi wk}{n}) \\
&= C_k\cdot e^{-\imath\pi\frac{k}{n}w}.
\end{aligned}
$$

$\square$

Now we prove Theorem 3.

**Theorem 3.** *Let $n < \infty$ be a finite positive integer. For any $\sigma \in (0, \infty]$ and $w, l \in [0, n]$, the derivatives of the Gaudi mask $\tilde{\boldsymbol{m}}^{\sigma}_{(w,l)}$ with respect to $w$ and $l$ are bounded almost everywhere:*

$$
\left\|\frac{\partial\tilde{\boldsymbol{m}}^{\sigma}_{(w,l)}}{\partial w}\right\|_2 < \infty, \quad \left\|\frac{\partial\tilde{\boldsymbol{m}}^{\sigma}_{(w,l)}}{\partial l}\right\|_2 < \infty.
$$

*Proof.* Here we use Parseval's theorem again:

$$
\begin{aligned}
\left\| \frac{\partial \tilde{\boldsymbol{m}}_{(w,l)}^{\sigma}}{\partial w} \right\|_2^2 &= \frac{1}{n} \sum_{k=0}^{n-1} \left| \frac{\partial}{\partial w} \mathrm{d}_w^{\sigma}[k] e^{-2\pi\imath \frac{k}{n} l} \right| \left| \frac{\partial}{\partial w} \mathrm{d}_w^{\sigma}[k] e^{-2\pi\imath \frac{k}{n} l} \right|^* \\
&= \frac{1}{n} \sum_{k=0}^{n-1} \left| e^{-2\pi\imath \frac{k}{n} w} \cdot e^{\pi\imath \frac{k}{n}} \cdot e^{-\frac{k^2}{2\sigma^2}} \cdot \frac{1}{\operatorname{sinc} \frac{k}{n}} e^{-2\pi\imath \frac{k}{n} l} \right|^2 \\
&= \frac{1}{n} \sum_{k=0}^{n-1} \left| e^{-\frac{k^2}{2\sigma^2}} \cdot \frac{1}{\operatorname{sinc} \frac{k}{n}} \right|^2 \\
&\leq \frac{1}{n} \sum_{k=0}^{n-1} \left| \frac{1}{\operatorname{sinc} \frac{n-1}{n}} \right|^2 \\
&< \infty,
\end{aligned}
$$

where the first inequality used the fact that $e^{-\frac{k^2}{2\sigma^2}} \leq 1$ and $\frac{1}{\operatorname{sinc} \frac{k}{n}} \leq \frac{1}{\operatorname{sinc} \frac{n-1}{n}}$ for $k = 0, 1, \ldots, n-1$. Similarly, the bound on the norm of the derivative with respect to the location parameter can be also derived as below:

$$
\begin{aligned}
\left\| \frac{\partial \tilde{\boldsymbol{m}}_{(w,l)}^{\sigma}}{\partial l} \right\|_2^2 &= \frac{1}{n} \sum_{k=0}^{n-1} \left| \mathrm{d}_w^{\sigma}[k] \frac{\partial}{\partial l} e^{-2\pi\imath \frac{k}{n} l} \right| \left| \mathrm{d}_w^{\sigma}[k] \frac{\partial}{\partial l} e^{-2\pi\imath \frac{k}{n} l} \right|^* \\
&= \frac{1}{n} \sum_{k=0}^{n-1} \left| \frac{\operatorname{sinc} \frac{wk}{n}}{\operatorname{sinc} \frac{k}{n}} e^{-\frac{k^2}{2\sigma^2}} e^{-\frac{2\pi\imath(1-w)}{n}} (-2\pi\imath \frac{k}{n}) e^{-2\pi\imath \frac{k}{n} l} \right|^2 \\
&\leq \frac{1}{n} \sum_{k=0}^{n-1} \left| \frac{1}{\operatorname{sinc} \frac{n-1}{n}} 2\pi \frac{n-1}{n} \right|^2 \\
&< \infty.
\end{aligned}
$$

$\square$

### A.1.6 PROOF OF COROLLARY 4

Before proving Corollary 4, we introduce another Corollary of Lemma 7.

**Corollary 8.** *For any $\sigma \in (0, \infty]$ and $w, l \in [0, n]$, the derivatives of the Gaudi mask $\tilde{\boldsymbol{m}}_{(w,l)}^{\sigma}$ with respect to $w$ and $l$ are given as follows:*

$$
\begin{aligned}
\frac{\partial \tilde{m}_{(w,l)}^{\sigma}[k]}{\partial w} &= \frac{1}{n} \sum_{k=0}^{n-1} F_{jk}^* \cdot e^{-2\pi\imath \frac{k}{n}(w+l)} \cdot e^{\pi\imath \frac{k}{n}} \cdot e^{-\frac{k^2}{2\sigma^2}} \cdot \frac{1}{\operatorname{sinc} \frac{k}{n}}, \\
\frac{\partial \tilde{m}_{(w,l)}^{\sigma}[k]}{\partial l} &= \frac{1}{n} \sum_{k=0}^{n-1} F_{jk}^* \cdot \left( -2\pi\imath \frac{k}{n} \right) \cdot e^{-2\pi\imath \frac{k}{n} l} \cdot \mathrm{d}_w^{\sigma}[k],
\end{aligned}
$$

*where $F_{jk}^* = e^{2\pi\imath j \frac{k}{n}}$ is the $(j, k)$th element of the inverse discrete Fourier transform (IDFT) matrix.*

*Proof.* By linearity of differentiation and Lemma 7

$$
\begin{aligned}
\frac{\partial}{\partial w}\tilde{m}^{\sigma}_{(w,l)}[k] &= \frac{\partial}{\partial w}\frac{1}{n}\sum_{k=0}^{n-1}F^{*}_{jk}\cdot e^{-2\pi\imath\frac{k}{n}l}\cdot \mathrm{d}^{\sigma}_{w}[k]\\
&= \frac{1}{n}\sum_{k=0}^{n-1}F^{*}_{jk}\cdot e^{-2\pi\imath\frac{k}{n}l}\cdot\frac{\partial}{\partial w}\mathrm{d}^{\sigma}_{w}[k]\\
&= \frac{1}{n}\sum_{k=0}^{n-1}F^{*}_{jk}\cdot e^{-2\pi\imath\frac{k}{n}l}\cdot e^{-2\pi\imath\frac{k}{n}w}\cdot e^{\pi\imath\frac{k}{n}}\cdot e^{-\frac{k^2}{2\sigma^2}}\cdot\frac{1}{\operatorname{sinc}\frac{k}{n}}\\
&= \frac{1}{n}\sum_{k=0}^{n-1}F^{*}_{jk}\cdot e^{-2\pi\imath\frac{k}{n}(w+l)}\cdot e^{\pi\imath\frac{k}{n}}\cdot e^{-\frac{k^2}{2\sigma^2}}\cdot\frac{1}{\operatorname{sinc}\frac{k}{n}}.
\end{aligned}
$$

The derivative with respect to the location parameter is a direct consequence of the derivative of the exponential function. $\square$

**Corollary 4.** *For any $l \in [0,n]$ and $\sigma \in (0,\infty]$, the norm of the derivative of the Gaudi mask $\tilde{\boldsymbol{m}}^{\sigma}_{(w,l)}$ with respect to $w$ at $w = 0$ is well-defined and greater than zero, i.e., $0 < \left\|\frac{\partial\tilde{\boldsymbol{m}}^{\sigma}_{(w,l)}}{\partial w}\right\|_{2} < \infty$.*

*Proof.* From Corollary 8

$$
\frac{\partial\tilde{m}^{\sigma}_{(w,l)}[k]}{\partial w}\bigg|_{w=0} = \frac{1}{n}\sum_{k=0}^{n-1}F^{*}_{jk}\cdot e^{-2\pi\imath\frac{k}{n}l}\cdot e^{\pi\imath\frac{k}{n}}\cdot e^{-\frac{k^2}{2\sigma^2}}\cdot\frac{1}{\operatorname{sinc}\frac{k}{n}}.
$$

Since $e^{-2\pi\imath\frac{k}{n}l}\cdot e^{\pi\imath\frac{k}{n}}\cdot e^{-\frac{k^2}{2\sigma^2}}\cdot\frac{1}{\operatorname{sinc}\frac{k}{n}}$ is not zero for any $l \in [0,n]$ and $\sigma \in (0,\infty]$ and the IDFT matrix $\boldsymbol{F}^{*}$ is unitary (up to scaling factor), the derivative $\frac{\partial}{\partial w}\tilde{\boldsymbol{m}}^{\sigma}_{(w,l)} \neq \boldsymbol{0}$. $\square$

## A.2 INITIALIZATION ALGORITHM

Suppose an $n$-by-$n$ matrix $\boldsymbol{W}_{\text{init}}$ is given before the initialization step. We first initialize the structural parameters based on the *correlation* within the columns of $\boldsymbol{W}_{\text{init}}$.

Consider the Gram matrix $\boldsymbol{C} = \boldsymbol{W}^{T}_{\text{init}}\boldsymbol{W}_{\text{init}}$. $C_{ij} = \boldsymbol{W}^{T}_{:,i}\boldsymbol{W}_{:,j}$ is the inner product between the $i$th column and the $j$th column of $\boldsymbol{W}_{\text{init}}$. That is, the higher $|C_{ij}|$ is, the more correlated the columns are. Thus, for the width and the location parameter of the column of the first block, we pick a row of the Gram matrix $\boldsymbol{C}_{i_1,:}$ where $i_1$ is the index of the row which has the **largest** norm. Then we apply a smoothing filter (e.g., Gaussian filter with the standard deviation $\gamma > 0$) to the absolute values of the row, which is followed by binarizing the filtered output based on some threshold $\tau > 0$. Hence, for the $k$th block, it can be summarized as follows:

$$
b_{i_k}[j] = \begin{cases} 1 & c_{i_k}[j] \geq \tau \\ 0 & c_{i_k}[j] < \tau \end{cases}, \quad \boldsymbol{c}_{i_k} = \text{Filter}(|\boldsymbol{C}_{i_k,:}|, \gamma), \quad i_k = \text{argsort}(\{\|\boldsymbol{C}_{i,:}\|_2\}^{n}_{i=1})[k]. \tag{12}
$$

As a result, the binary vector $\boldsymbol{b}_i$ consists of the chunks of ones and zeros. We choose the longest chunk of ones and initialize the width and the location parameters $(w^{C}_k, l^{C}_k)$ correspondingly. For the row parameters of the $k$th block, the same procedure is repeated with the columns of $\boldsymbol{W}_{\text{init}}$ selected in the column parameter initialization step.

Next, the content parameters are set by the left and the right singular vectors of $\boldsymbol{W}_{\text{init}}$ scaled by the singular values, i.e., $\boldsymbol{U} \leftarrow \boldsymbol{U}_{\text{init}}\boldsymbol{S}^{1/2}_{\text{init}}$ and $\boldsymbol{V} \leftarrow \boldsymbol{S}^{1/2}_{\text{init}}\boldsymbol{V}^{T}$ where $\boldsymbol{W}_{\text{init}} = \boldsymbol{U}_{\text{init}}\boldsymbol{S}_{\text{init}}\boldsymbol{V}^{T}_{\text{init}}$.

Finally, we update the structural parameters and content parameters by minimizing the following objective function:

$$
\min_{\boldsymbol{U},\boldsymbol{V},\boldsymbol{\phi}} \left\|\boldsymbol{W}_{\text{init}} - \boldsymbol{W}^{\boldsymbol{\theta}}\right\|^{2}_{F} + \lambda\|\boldsymbol{w}\|_1 + \boldsymbol{1}_{\boldsymbol{w}\in\mathcal{I}^{n}_{n}} + \boldsymbol{1}_{l\in\mathcal{I}^{n}_{n-1}} \tag{13}
$$

We summarize the initialization steps in Algorithm 2.

---

**Algorithm 2** GBLR Parameter Initialization

1: $\boldsymbol{C} \leftarrow \boldsymbol{W}^T \boldsymbol{W}$                                      ▷ Target matrix $\boldsymbol{W} \in \mathbb{R}^{n \times n}$
2: $i_1, i_2, \ldots, i_n \leftarrow \mathrm{argsort}(\{\|\boldsymbol{C}_{i,:}\|_2\}_{i=1}^n)$
3: $b \leftarrow 0$
4: **for** $k = 1, \ldots, n$ **do**
5:      $\phi_k^C \leftarrow \texttt{get\_mask\_params}(|\boldsymbol{C}_{i_k,:}|, \gamma, \tau)$
6:      $\boldsymbol{R} = (\boldsymbol{W} \cdot \boldsymbol{m}_k^C)(\boldsymbol{W} \cdot \boldsymbol{m}_k^C)^T$
7:      $i_k' \leftarrow \mathrm{argmax}(\{\|\boldsymbol{R}_{i,:}\|_2\}_{i=1}^n)$
8:      $\phi_k^R \leftarrow \texttt{get\_mask\_params}(|\boldsymbol{R}_{i_k',:}^T|, \gamma, \tau)$
9:      $b \leftarrow b + \mathrm{nnz}(\boldsymbol{m}_k^c) + \mathrm{nnz}(\boldsymbol{m}_k^R)$
10:      **if** $b > s$ **then**
11:          $w_k^c \leftarrow 0, w_k^R \leftarrow 0$; **break**
12:      **end if**
13: **end for**
14: $\boldsymbol{U}, \boldsymbol{V} \leftarrow \mathrm{SVD}(\boldsymbol{W})$
15: Solve Eq. 13 by Alg. (1).

---

## A.3 RECTENGULAR GAUDI-GBLR MATRICES

When the number of rows $m$ is not equal to the number of columns $n$, the width and location parameters for rows are defined on $\mathcal{I}_m$ and $\mathcal{I}_{m-1}$, respectively, whereas the structural parameters on columns are defined on $\mathcal{I}_n$ and $\mathcal{I}_{n-1}$. Except for the index sets, the GBLR matrix on the rectangular case is defined exactly the same. Theorem 1 and 2 hold as well. The Gaudi mask is also defined in the same manner.

## A.4 GAUDI MASK WITH REAL-VALUED WIDTHS AND LOCATIONS

Let us consider the Gaudi mask in Eq. 6 with the continuous-valued widths and locations $w, l \in [0, n]$. Also, let us assume for now that $\sigma = \infty$, namely, no Gaussian smoothing is applied. On one hand, Eq. 6 still compiles the time-domain signal after IDFT. On the other hand, the output of the IDFT is not a boxcar mask anymore if at least one of $w$ and $l$ is not an integer. In Figure 7, we present the time-domain signal of three cases of the Gaudi mask on $n = 512$: 1) integer-valued width and locations with the smoothing parameter $\sigma = \infty$, namely, no Gaussian filter is applied in the Gaudi function; 2) non-integer-valued width and location with $\sigma = \infty$; and 3) non-integer-valued width and location with $\sigma = 100$. As illustrated in the figure, the non-integer-valued parameters generate many spiking errors (middle). Notably, however, the errors are alleviated when the smoothing parameter $\sigma = 100$ (right), which almost recovers the shape of the signal with the integer-valued parameters (left).

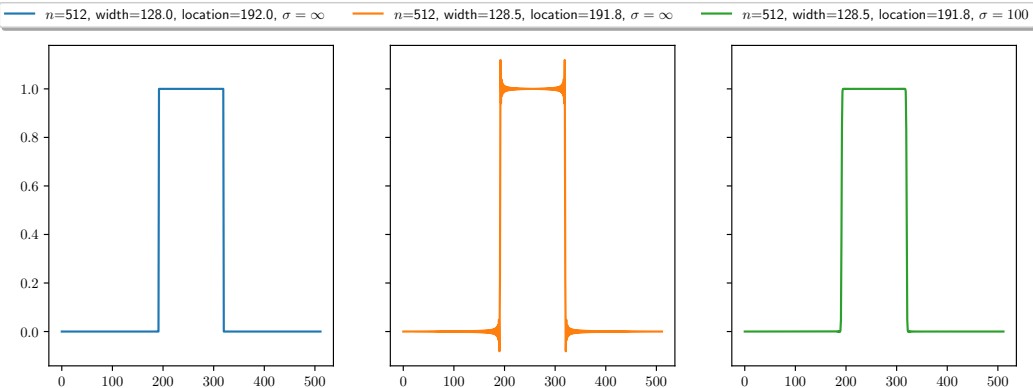

Figure 7: Gaudi Masks in time-domain with different width, location and $\sigma$ on $n = 512$. Left: integer width $w = 128$ and integer location $l = 192$ with $\sigma = \infty$, i.e., no Gaussian smoothing. Middle: non-integer width $w = 128.5$ and location $l = 191.8$ with $\sigma = \infty$. Due to the non-integer parameters, the time-domain signal contains spiking errors. Right: non-integer width $w = 128.5$ and location $l = 191.8$ with $\sigma = 100$. The Gaussian smoothing in the Gaudi function alleviates the spiking errors.

## A.5 MORE DETAILS ON PROXIMAL GRADIENT DESCENT ON GAUDI-GBLR PARAMETERS IN DEEP NEURAL NETWORKS

### A.5.1 DISCRETE STRUCTURAL PARAMETERS

Here we discuss solving Problem (8) using the discrete width and location parameters $w \in \mathcal{I}_n$ and $l \in \mathcal{I}_{n-1}$. To be specific, given real-valued widths $\boldsymbol{w} \in [0, n]^K$ and locations $\boldsymbol{l} \in [0, n]^K$ of a Gaudi-GBLR matrix, we apply a Straight Through Estimator (Hinton, 2012; Bengio et al., 2013; Zhou et al., 2016) to $\boldsymbol{w}$ and $\boldsymbol{l}$ *before* generating the masks. Then, we project each value of $\boldsymbol{w}$ and $\boldsymbol{l}$ to $[0, n]$ *after* the PGD step. We find that this setting is useful when the Gaussian smoothing parameter $\sigma$ of the Gaudi function is very high or infinite, since the non-integer parameters generate spiking errors as discussed in Appendix A.4.

### A.5.2 MULTIPLE GAUDI-GBLR MATRICES

For a DNN with $T$ weight matrices, we suggest adjusting the shrinkage parameter $\lambda$ in eq. 9 based on the predefined computational cost *budget B* for the DNN. For example, one can set $\lambda = 0$ if the sum of average widths of the weight matrices $\bar{w}_{\text{sum}} = \bar{w}(\boldsymbol{W}_1) + \bar{w}(\boldsymbol{W}_2) + \cdots + \bar{w}(\boldsymbol{W}_T)$ is below $B$, and use $\lambda = \lambda_0 > 0$ if $\bar{w}_{\text{sum}}$ is above $B$ where $\lambda_0$ is given as a hyperparameter by user. In our experiments, this strategy effectively (not necessarily equally) distributes the computational cost to each matrix to meet the overall cost budget constraint during the learning process.

## A.6 IMPACT OF SVD

**SVD.** Since our method involves Singular Vector Decomposition (SVD) during initialization, the time complexity of the SVD is discussed in this section. The impact caused by SVD is negligible in the training process. The SVD operation performed during initialization decomposes at most $4096 \times 1024$-sized matrix, which takes less than a second on commercial CPUs or GPUs, whereas the overall training or fine-tuning time ranges from a few hours to tens of hours. Also, the SVD does not occur during the inference stage, so it does not become a bottleneck in any sense once the network is trained.

## A.7 EXTENDED EXPERIMENTAL RESULTS AND DETAILS

### A.7.1 FINE-TUNING

In fine-tuning experiments, we use the pre-trained ViT-Base (Dosovitskiy et al., 2020). For each pre-trained weight on Query, Key, Value and MLPs, we assign the Gaudi-GBLR matrix where the structural and content parameters are initialized by Algorithm 2. We use the smoothing parameter of the filter $\gamma = 1$ and the threshold to obtain the binary mask $\tau = 0.98 \cdot \max(\boldsymbol{c}_{i_k})$ in Eq. 12. During the fine-tuning, we use $\sigma = \infty$ in Eq. 6 Then, all parameters are updated by SGD with AdamW(Loshchilov & Hutter, 2017) optimizers.

For the Block Low-Rank matrices, we use Monarch (Dao et al., 2022) implementation. We project the pre-trained matrix by minimizing the Frobenius norm between the pretrained one and the Monarch matrix by gradient descent for 1000 steps with the learning rate of $0.001$.

The initialization learning rate for the structural parameters was set to $0.0025$ and decayed to $0.000025$. We trained the networks for 35 epochs with the learning rate of $0.0001$ for all parameters, including the structural parameters of the Gaudi-GBLR matrices.

For the WikiText103 experiments, we used the publicly available GPT-2 model[1] for the pre-trained weights.

### A.7.2 TRAINING FROM SCRATCH

For the Gaudi-GBLR matrices, we initialize location parameters to zero and width parameters to the low rank + block sparse structure. The content parameters as well as other weights are initialized randomly.

During the training with Proximal Gradient Descent in Eq. 9, as in Appendix A.5, the shrinkage parameter $\lambda$ is set to zero if the average width of the Gaudi-GBLR matrices of the DNN is below the predefined budget, and recovers the predefined value $\lambda_0$ otherwise.

Also, we anneal $\sigma = 1$ to 100 during the training process. $\sigma$ is fixed to 1 for 5 warm-up epochs, then linearly increases to 100 for 295 epochs. The hyperparameters used in the ImageNet experiment is presented in Table 4.

Table 4: Hyperparameters used in ImageNet experiments using ViT-Base for training from scratch.

| Model | Learning Rate | | Shrinkage | | $\sigma$ (init/final) | Epochs | Weight Decay | Drop Path Rate |
|---|---|---|---|---|---|---|---|---|
| | Structural Params | Content Params | $\lambda_0$ | Target Width | | | | |
| ViT-Base | N/A | 0.001 | N/A | N/A | N/A | 310 | 0.05 | 0.1 |
| ViT-Base Gaudi-GBLR | 0.001 | 0.001 | 0.04 | $0.12 \cdot 768$ | 1.0/100.0 | 310 | 0.05 | 0.0 |

## A.8 EXTENDED EXPERIMENTAL RESULTS

### A.8.1 GBLR MATRIX FOR CONVOLUTION WEIGHTS.

We compare the ImageNet accuracy of ResNet18 (He et al., 2016) with GBLR weight matrices to the Automatic Layer-wise Decomposition Selector (ALDS) (Liebenwein et al., 2021), a similar rank-based layer-wise compression method. Following Liebenwein et al. (2021), we consider the $(d, c, k_1, k_2)$-sized weight tensor of a convolution layer as $d \times (ck_1k_2)$-sized matrix, where $d, c, k_1, k_2$ are the number of output channels, input channels, and the widths and heights of the kernels, respectively. In this manner, we converted the weights of the convolution layers of the ResNet18 model to GBLR matrices, and retrained the model. Table 5 shows the accuracy and the relative FLOPs of the ResNet18 model with dense weights, low-rank weights found by one-shot ALDS, and GBLR weights found by our method. Even without any supervision on the structural properties of the convolution kernels, GBLR outperforms ALDS low-rank weights in terms of both accuracy and complexity.

---

[1]https://huggingface.co/Graphcore/gpt2-wikitext-103

Table 5: ImageNet Accuracy on ResNet18.

| Weight Type | Accuracy | GFLOPs |
|---|---|---|
| Dense | 69.62 | 1.82 |
| ALDS (Liebenwein et al., 2021) | 69.22 | 1.03 |
| GBLR (ours) | **69.31** | **1.01** |

### A.8.2 GENERALIZATION GAP OF DNNs WITH GBLR WEIGHTS

In this section, we estimate the generalization gap of the DNNs with GBLR weights. We used the ViT-Base model discussed in Section 4.3 which is trained on ImageNet from scratch. The accuracy of the dense and GBLR models on the training and validation sets are evaluated and presented in Table 6. Notably, the training-validation accuracy difference of the GBLR model is smaller than that of the dense model. The comparable validation accuracy despite the lower training accuracy provides reasonable empirical evidence that the DNNs with GBLR matrices may exhibit smaller generalization errors than dense models.

Table 6: Generalization Gap of Dense and GBLR ViT-Base Models.

| Weight Type | Training Acc. (%) | Validation Acc. ($|\Delta|$) (%) | Training Loss | Validation Loss ($|\Delta|$) |
|---|---|---|---|---|
| Dense | 99.03 | 78.57 (20.46) | 0.1640 | 1.0639 (0.8999) |
| Gaudi-GBLR | 94.93 | 78.51 (16.42) | 0.3260 | 0.9917 (0.6657) |

