# OpenReview forum: "Differentiable Learning of Generalized Structured Matrices for Efficient Deep Neural Networks"
_ICLR.cc/2024/Conference — ICLR 2024 poster_

### Official Review · Reviewer_wuLi · 2023-10-31

**Soundness:** 3 good
**Presentation:** 3 good
**Contribution:** 3 good
**Rating:** 6
**Confidence:** 5

**Summary:**

The paper presents a new compressed matrix format (parametrization) called GBLR and an optimization method (involving proximal gradient descent, homotopy, and STE) to learn the parameters of the format to make NNs faster and smaller. The proposed parametrization can be understood as a juxtaposition of low-rank matrices of various shapes that needs to be appropriately padded. Such a parametrization is very flexible as contains regular low-rank matrices (if the shape of the blocks are same as the shape of the entire matrix), block low rank matrices, and block sparse. To be more specific, every low-rank submatrix  is represented as a sum of rank-1 matrices; thus, for a given subblock of rank $k$ parameters include shape of submatrix ($w$,$h$), location within orgininal matrix (i,j), and actual values that are $k$ rank-1 matrices of $w \times h$ stored as $w\times 1$ and $1 \times h$ updates.

Such a parametrization has many non-differentiable parameters, and to make it amenable to SGD based solvers the authors propose several modifications: instead of storing compact $w\times 1$ and $1\times h$ rank-1 matrices, the parametrization now involves the whole $n\times 1$ and $1\times n$ vectors that are appropirately masked with vector $m$. The mask presents a boxcar filter which is non-differentiable itself, but author approximate it via Gaudi mask in frequency domain with gaussian kernel (with variance $\sigma^2$); Gaudi mask converges to boxcar with $\sigma \to \infty$ which needs to be driven during training (hence a homotopy). Also, during the training the width $w$, height $h$, location $i,h$, are kept real-valued (for SGD) but apply straight-through estimation for actual matrix recreation. And finally, to drive matrix rank selection, authors propose a cost based selection via an $\ell_1$ penalty (controlled by $\lambda$) on certain structure parameters (since the exact equation is not given, I'm guessing on every rank-1 matrix?).

As for training of the compressed models, the authors plug in GBLR parametrized matrix instead of original weight matrices of nns (the initial values are obtained via an algorithm in A.2), choose appropriate $\lambda$ (to control for rank), and train or finetune end-to-end on a dataset. The results clearly show that such a scheme gets a better compression-accuracy tradeoff.

**Strengths:**

The paper proposed a new matrix parametrization that includes many other compressed forms (low rank, block low-rank, and block sparse) as a subset. The parametrization allows to get better error-compression tradeoffs.

The presentation of the method is very thorough and includes many small details (except for some, see questions) that is definitely a plus for reproducibility.

**Weaknesses:**

I find two minor weaknesses of the paper, literature review and comparison to other methods.

Although I understand that the focus of paper was compression of transformer based models, it seems many relevant low-rank and tensor-decomposition based methods that were used for compression of other networks (CNNs) were left out. Many of those left out papers share similar ideas (e.g., how to parametrize wrt rank) that need to be included. Some of the missed out works include:
- Factorized Higher-Order CNNs with an Application to Spatio-Temporal Emotion Estimation
- Low-rank Compression of Neural Nets: Learning the Rank of Each Layer
- Coordinating filters for faster deep neural networks
- Constrained optimization based low-rank approximation of deep neural networks
- Compressing Neural Networks: Towards Determining the Optimal Layer-wise Decomposition

but there are many others (you can find others by looking within those)

Comparison to the relevant baselines. It seems the baselines authors choose are very simple (e.g., using a fixed rank for low-rank compression), and can be considerably strengthened if wanted. Thus I'm asking authors to include stronger baselines (low rank with rank selection, tensor decomposition methods) to compare.

**Questions:**

1. Can you please provide the exact from of $\ell_1$ penalty used in eq.8? How does FLOPs/parameter counts being represented as F$\ell_1$ penalty?
2. What is the value of $\lambda$ used in experiments? How to choose it properly?
3. What was the scheme used for $sigma$? The paper says that it was "gradually increased to 100", but having exact details is preferred.
4. Certain computations (Gaudi function) happens in frequency domain that involves DFT and IDFT; how expensive is this operations wrt single training step? No slowdown, 0.75x slowdown, etc..
5. Please included a more detailed literature review.
6. Please strengthen the baselines.
7. I am wondering why section 3.3 discusses the application on the basis of two layer MLP "for ease of understanding"? Wouldn't an example on a single layer be much simpler? BTW, there is a typo in this section, multi-layer perception => multi-layer perceptron

---

> ### Author Response · Authors · 2023-11-18
>
> We appreciate the constructive suggestions! We revised and extended the manuscript based on your comments.
>
> **Q1:** It seems many relevant low-rank and tensor-decomposition based methods that were used for compression of other networks (CNNs) were left out. Many of those left out papers share similar ideas (e.g., how to parameterize wrt rank) that need to be included.
>
>
> **A1:** We carefully went through the suggested methods used in CNNs.
> We added a new comparison for the ImageNet accuracy of ResNet18 with GBLR weight matrices vs. the Automatic Layer-wise Decomposition Selector (ALDS) [1], which is a similar rank-based layer-wise compression method.
> Following [1], we consider the $(d, c, k_1, k_2)$-sized weight tensor of a convolution layer as a $d \times (c k_1 k_2)$-sized matrix, where $d, c, k_1, k_2$ are the number of output channels, input channels, and the widths and heights of the kernels, respectively.
> In this manner, we converted the weights of the convolution layers of the ResNet18 model to GBLR matrices, and retrained the model.
> Table 1 shows the accuracy and the relative FLOPs of the ResNet18 model with dense weights, low-rank weights found by one-shot ALDS [1], and GBLR weights found by our method.
> Even without any supervision on the structural properties of the convolution kernels, GBLR outperforms ALDS low-rank weights in terms of both accuracy and complexity.
>
> **Table 1.**
> | ResNet18 Weight Matrix | Accuracy (\%) | Relative FLOPs (\%) |
> |------------------------|---------------|---------------------|
> | Dense                  |         69.62 |                 100 |
> | ALDS [1]               |         69.22 |               56.49 |
> | GBLR (ours)            |     **69.31** |           **55.62** |
>
>
> **Q2:** Can you please provide the exact form of $\ell_1$ penalty used in eq.8? How does FLOPs/parameter counts being represented as $\ell_1$ penalty?
>
> **A2:** The FLOPs / parameter count is directly represented by the sum of the width parameters as in Eq. 4 (restated below):
> $$
> \mathrm{FLOPs}=\sum_{k=1}^K (w_k^R + w_k^C),
> $$
> which is the $\ell_1$ norm of the vector containing width parameters $\boldsymbol{w}=\{w_1^R, w_1^C,\ldots,w_K^R,w_K^C\}$.
> Hence, minimizing the $\ell_1$ norm of this width parameter vector $\boldsymbol{w}$ of a GBLR matrix reduces the FLOPs for matrix-vector product.
>
>
>
> **Q3:** What is the value of $\lambda$ used in experiments? How to choose it properly?
>
> **A3:** We used $\lambda=0.02-0.04$ for the ViTs, $\lambda=0.1-0.2$ for the MLP-Mixers, and $\lambda=0.001-0.005$ for ResNet-18. The range of the optimal choice for the $\lambda$ hyperparameter depends on the model, learning rate, and dataset. We used a grid search enumerating the search space to find the $\lambda$ hyperparameter in our experiments. Automatically selecting an optimal $\lambda$ from the budget is left as a topic for future work.
>
> **Q4:** What was the scheme used for $\sigma$? The paper says that it was "gradually increased to 100", but having exact details is preferred.
>
>
> **A4:** We initially set $\sigma$ to 1. At each epoch, we start increasing $\sigma$ linearly to 100 until the end of the training process, except during the learning rate warm-up and cool-down steps. For example, if the training process consists of 110 epochs including 5 warm-up epochs at the beginning and 10 cool-down epochs at the end, $\sigma$ remains at 1 until epoch 5 and then increases linearly to 100 until epoch 100.
> We provided the $\sigma$ scheduling used in our experiment in Appendix A.7.2.
>
> **Q5:** Certain computations (Gaudi function) happens in frequency domain that involves DFT and IDFT; how expensive is this operations wrt single training step? No slowdown, 0.75x slowdown, etc..
>
> **A5:** FFT/IFFT does not impose significant overhead in our methodology. Let us consider ViT-Large for example. At each fully-connected layer, 1-D Inverse Fast Fourier Transform (IFFT) is performed over a $4096 \times 1024$-sized weight matrix, which requires $4096 \times 2.5 \times 1024 \log_2 1024 = 104,857,600 \approx 105\times 10^6$ FLOPs.
>
> In contrast, the matrix-matrix multiplication (MMM) between a $(B, N, 4096)$-sized input tensor and a $4096\times 1024$-sized weight matrix requires $B\times N \times 4096 \times 1024$ FLOPs where $B$ and $N$ are the batch size and sequence length, respectively. For $B=128$ and $N=197$, the number of FLOPs for MMM is $106\times 10^9$, which is three orders of magnitude larger than the number of FLOPs required for IFFT.
>
>
>
> **Reference**
>
> [1] Liebenwein, Lucas, et al. "Compressing neural networks: Towards determining the optimal layer-wise decomposition." Advances in Neural Information Processing Systems 34 (2021): 5328-5344.

---

> > ### Author Response · Authors · 2023-11-18
> >
> > **Q6:** I am wondering why section 3.3 discusses the application on the basis of two layer MLP "for ease of understanding"? Wouldn't an example on a single layer be much simpler?
> >
> > **A6:** In order to illustrate our proposed method, we chose a 'multi'-layer perceptron. We opted for a two-layer MLP, as the simplest form of a neural network. Although a single-layer model is much simpler, we thought it might potentially confuse readers as it is essentially a linear model, not a neural network. Furthermore, a single-layer model with a scalar output does not effectively capture our method because, in such a model, the weight is a vector and not a matrix.

---

> > > ### Author Response · Authors · 2023-11-23
> > > **Thank you for the constructive review.**
> > >
> > > We hope our response has addressed all of your questions. We improved the quality of our manuscript significantly based on your invaluable input, and we sincerely appreciate your effort.

---

### Official Review · Reviewer_f3RM · 2023-11-04

**Soundness:** 2 fair
**Presentation:** 2 fair
**Contribution:** 2 fair
**Rating:** 5
**Confidence:** 3

**Summary:**

This paper proposes a generalized and differentiable method to learn efficient structures of weight matrices. Moreover, the authors present an effective initialization technique for the proposed method. Some experimental results show the performance of the proposed method.

**Strengths:**

1. This paper proposes a generalized and differentiable method to learn efficient structures of weight matrices.
2. Moreover, the authors present an effective initialization technique for the proposed method.
3. Some experimental results show the performance of the proposed method.

**Weaknesses:**

Although the paper is theoretically and experimental sound, there are still some questions need to be discussed in this paper:
1.	In Algorithm 1, what’s AdamW(.), as well as clip?
2.	In Eq. (8), what’s the variable, w or \theta?
3.	The advantage of the proposed method against existing methods is not clear.
4.	The parameter initialization for the proposed method needs to perform SVD. Thus, the authors should analyze the computational complexity.
5.	The experimental results are not convincing. The authors should compare the performance of the proposed algorithm and more methods on more models and datasets.
6.	The English language in this paper needs to be improved.

**Questions:**

Although the paper is theoretically and experimental sound, there are still some questions need to be discussed in this paper:
1.	In Algorithm 1, what’s AdamW(.), as well as clip?
2.	In Eq. (8), what’s the variable, w or \theta?
3.	The advantage of the proposed method against existing methods is not clear.
4.	The parameter initialization for the proposed method needs to perform SVD. Thus, the authors should analyze the computational complexity.
5.	The experimental results are not convincing. The authors should compare the performance of the proposed algorithm and more methods on more models and datasets.
6.	The English language in this paper needs to be improved.

---

> ### Author Response · Authors · 2023-11-18
>
> Thank you for the comprehensive suggestions! The below provides answers to some of your questions.
>
> **Q1:** In Algorithm 1, what’s AdamW(.), as well as clip?
>
> **A1:** $\mathrm{AdamW}(\cdot )$ stands for the AdamW optimizer [1]. Line 5 in Algorithm 1 ($p \gets p - \eta \cdot \mathrm{AdamW}(\nabla \ell)$) corresponds to the conventional parameter update to minimize the loss $\ell$ by using the AdamW optimizer.
>
> The function $\mathrm{clip}(\mathbf{w}, 0, n)$ clamps each element of $\mathbf{w}$ to the interval $[0, n]$, which is the domain of the width parameter. We included this operation in Algorithm 1 to ensure that the width parameter stays within $[0,n]$ after the gradient descent.
>
> We have added a detailed explanation of Algorithm 1 in Section 3.3.
>
> **Q2:** In Eq. (8), what’s the variable, w or $\theta$?
>
>
> **A2:** In Eq. (8), the variable $\theta$ denotes the set of the structural and content parameters, namely, $\theta=(\boldsymbol{\phi}, \boldsymbol{U}, \boldsymbol{V}, \sigma)$, as stated above Eq. (7).
> The variable $\boldsymbol{w}$ denotes the width parameters $\boldsymbol{w}=\{w_1^R, w_1^C,\ldots,w_K^R,w_K^C\}$ of $\boldsymbol{W}^{\boldsymbol{\theta}}$. We updated Section 3.3 to clarify the definition of $\boldsymbol{w}$.
>
> **Q3:** The advantage of the proposed method against existing methods is not clear.
>
> **A3:** The key advantage of our method lies in its ability to *learn the structural parameters of weight matrices* by stochastic gradient descent, overcoming the non-differentiable nature of typical matrix structures --e.g., non-zero locations of the sparse matrix or the rank of the low-rank matrix. Prior approaches used hand-crafted structures because of this issue. Our approach allows treating structural parameters (widths and locations of the masks in Eq. 2) as additional trainable parameters to automatically identify/learn efficient structures in practice.
> This enables the end-to-end, layer-wise learning of the optimal structured matrix for each weight matrix given the computational budget for each layer.
>
> Most existing rank-based compression methods are confined to either a single type of structured matrix (e.g., low-rank or block low-rank) [2,3] relying on a heuristic rule for budget allocation across layers [3,4], or segregating the structural parameter optimization from the training process of the other parameters [2,4].
> This is primarily because, in those prior approaches, the structural formats are defined in a non-differentiable and combinatorial manner.
>
> Using the learned structures and parameters from our approach generally provides higher accuracy for the same complexity (or comparable accuracy with lower complexity) compared to the model in prior works.
>
> **Q4:** The parameter initialization for the proposed method needs to perform SVD. Thus, the authors should analyze the computational complexity.
>
> **A4:** Please note that our objective is to minimize the inference complexity. And the SVD overhead during training is negligible. The SVD during initialization takes less than one minute, whereas the overall training or fine-tuning time ranges from a few hours to tens of hours. Also, the SVD is unnecessary during the inference stage once the network is trained. We have included an analysis of the impact of SVD on training time in the Appendix.
>
> **References**
>
> [1] Loshchilov, Ilya, and Frank Hutter. "Decoupled Weight Decay Regularization." International Conference on Learning Representations. 2018.
>
> [2] Liebenwein, Lucas, et al. "Compressing neural networks: Towards determining the optimal layer-wise decomposition." Advances in Neural Information Processing Systems 34 (2021): 5328-5344.
>
> [3] Chen, Beidi, et al. "Pixelated Butterfly: Simple and Efficient Sparse training for Neural Network Models." International Conference on Learning Representations. 2021.
>
> [4] Dao, Tri, et al. "Monarch: Expressive structured matrices for efficient and accurate training." International Conference on Machine Learning. PMLR, 2022.

---

> > ### Author Response · Authors · 2023-11-18
> >
> > **Q5:** The authors should compare the performance of the proposed algorithm and more methods on more models and datasets.
> >
> >
> > **A5:** We have extended our experimental results in two ways:
> > 1. Additional evaluation of a new model (CNN) against a stronger baseline (ALDS [2]),
> > 2. Additional evaluation on a language model using a text generation task on an additional dataset (WikiText-103) using GPT2.
> > We have included additional discussions and results to the manuscript as summarized below:
> >
> >
> > **CNN Results.**
> > We compare the ImageNet accuracy of ResNet18 with GBLR weight matrices to the Automatic Layer-wise Decomposition Selector (ALDS) [2], a similar rank-based layer-wise compression method.
> > Following [2], we consider the $(d, c, k_1, k_2)$-sized weight tensor of a convolution layer as $d \times (c k_1 k_2)$-sized matrix, where $d, c, k_1, k_2$ are the number of output channels, input channels, and the widths and heights of the kernels, respectively.
> > In this manner, we converted the weights of the convolution layers of the ResNet18 model to GBLR matrices, and retrained the model.
> > Table 1 shows the accuracy and the relative FLOPs of ResNet18 models with dense weights, low-rank weights found by ALDS [2], and GBLR weights found by our method.
> > Even without any supervision on the structural properties of the convolution kernels, GBLR moderately outperforms ALDS low-rank weights in terms of both accuracy and complexity.
> >
> >
> > **Table 1.**
> > | ResNet18 Weight Matrix | Accuracy (\%) | Relative FLOPs (\%) |
> > |------------------------|---------------|---------------------|
> > | Dense                  |         69.62 |                 100 |
> > | ALDS [2]               |         69.22 |               56.49 |
> > | GBLR (ours)            |     **69.31** |           **55.62** |
> >
> > **Language Model Results.**
> > In this experiment, we fine-tuned the pre-trained GPT2 model using the WikiText-103 dataset. Table 2 presents the perplexity (the lower the better) of the GPT2 model with Dense, Low-Rank, Monarch [4], and Gaudi-GBLR matrices. Our proposed method achieves the lowest perplexity, outperforming the dense baseline and utilizing the smallest number of FLOPs. We generated more data points from different relative FLOPs (35\% and 52.5\%) and included them in the manuscript alongside the result presented in Table 2.
> >
> >
> > **Table 2.**
> > | GPT2 Weight Matrix | Perplexity ($\downarrow$) | Relative FLOPs |
> > |--------------------|--------------------------:|---------------:|
> > | Dense              |                     19.36 |          100\% |
> > | Low Rank           |                     19.48 |        43.75\% |
> > | Monarch [4]        |                     20.56 |        43.75\% |
> > | Gaudi-GBLR (ours)  |                 **19.24** |     **43.7\%** |
> >
> >
> > **References**
> >
> > [1] Loshchilov, Ilya, and Frank Hutter. "Decoupled Weight Decay Regularization." International Conference on Learning Representations. 2018.
> >
> > [2] Liebenwein, Lucas, et al. "Compressing neural networks: Towards determining the optimal layer-wise decomposition." Advances in Neural Information Processing Systems 34 (2021): 5328-5344.
> >
> > [3] Chen, Beidi, et al. "Pixelated Butterfly: Simple and Efficient Sparse training for Neural Network Models." International Conference on Learning Representations. 2021.
> >
> > [4] Dao, Tri, et al. "Monarch: Expressive structured matrices for efficient and accurate training." International Conference on Machine Learning. PMLR, 2022.

---

> > > ### Author Response · Authors · 2023-11-23
> > > **Thank you for the constructive review.**
> > >
> > > We hope our response has addressed all of your questions. We improved the quality of our manuscript significantly based on your invaluable input, and we sincerely appreciate your effort.

---

### Official Review · Reviewer_g58u · 2023-11-08

**Soundness:** 3 good
**Presentation:** 3 good
**Contribution:** 2 fair
**Rating:** 6
**Confidence:** 3

**Summary:**

The authors introduce a Generalized Block-low-rank  (GBLR) matrix format to construct computationally efficient structures of weight matrices. They also introduce Gaussian-Dirichlet (Gaudi) function to make the structural parameters differentiable and provide an algorithm to learn neural networks with Gaudi-GBLR weight matrices.

**Strengths:**

The proposed GBLR format includes existing important matrix structures. Also, the authors provide a method to make the structural parameters of weight matrices learnable. The idea is interesting and relevant to the community.

**Weaknesses:**

Providing theoretical investigations of the neural networks learned by the proposed method can improve the quality of the paper. Since the weight matrices are forced to be sparse, I think we need a different analysis from existing analysis for the dense weight matrices. For example, do you have any explanation about the representation power and generalization property of the networks with GBLR weight matrices?

**Questions:**

As I also mentioned in the weakness part, how does the GBLR weight matrices affect the generalization property or the complexity of the neural network?

---

> ### Author Response · Authors · 2023-11-18
>
> Thank you for the insightful feedback!
>
> **Q1:** How does the GBLR weight matrices affect the generalization property or the complexity of the neural network?
>
> **A1:** We hypothesize based on empirical observations that the GBLR weight matrices reduce the generalization gap of neural networks.
> We observed that when the weights of DNNs are initialized to the GBLR matrices and trained from scratch, the *training* accuracy is generally lower than that of the dense model, whereas the *validation accuracy* is similar. For instance, Table 1 shows the ImageNet accuracy of dense and GBLR ViT-Base models.
>
> **Table 1.**
>
> | ViT-Base   | Training Accuracy (\%) | Validation Accuracy ($\|\Delta\|$) (\%) | Training Loss | Validation Loss ($\|\Delta\|$) |
> |------------|-----------------------:|----------------------------------------:|--------------:|-------------------------------:|
> | Dense      |                  99.03 |                           78.57 (20.46) |        0.1640 |                1.0639 (0.8999) |
> | Gaudi-GBLR |                  94.93 |                           78.51 (16.42) |        0.3260 |                0.9917 (0.6657) |
>
> The comparable validation accuracy despite the lower training accuracy provides reasonable empirical evidence that the DNNs with GBLR matrices may exhibit smaller generalization errors than the dense models. We added this discussion to the paper.
>
>
> However, given the low-rankness and sparsity of GBLR matrices, we agree with the reviewer that we need a different technique to fully understand the generalization and approximation bounds of the DNNs with GBLR matrices.
> Due to time constraints, unfortunately, we could not conclusively determine the theoretical generalizability of the GBLR matrices.
> Nonetheless, we believe the generalized structured format has potential to serve as a good tool for a better understanding of the generalization bounds of DNNs using the low-rank matrices [1,2,3] and the approximation bounds of DNNs with sparse matrices [4,5]. Conceptually, they are based on the low-dimensional representations of large and complex functions. Since our method expands the low-dimensional matrix representations from the hand-designed models to a general parametric space, we are eager to study the GBLR matrix theoretically in future work.
>
>
>
> **References**
>
> [1] Arora, Sanjeev, et al. "Stronger generalization bounds for deep nets via a compression approach." International Conference on Machine Learning. PMLR, 2018.
>
> [2] Suzuki, Taiji, Hiroshi Abe, and Tomoaki Nishimura. "Compression based bound for non-compressed network: unified generalization error analysis of large compressible deep neural network." International Conference on Learning Representations. 2020.
>
> [3] Baykal, Cenk, et al. "Data-Dependent Coresets for Compressing Neural Networks with Applications to Generalization Bounds." International Conference on Learning Representations. 2018.
>
> [4] Klusowski, Jason M., and Andrew R. Barron. "Approximation by combinations of ReLU and squared ReLU ridge functions with $\ell^ 1$ and $\ell^ 0$ controls." IEEE Transactions on Information Theory 64.12 (2018): 7649-7656.
>
> [5] Domingo-Enrich, Carles, and Youssef Mroueh. "Tighter Sparse Approximation Bounds for ReLU Neural Networks." International Conference on Learning Representations. 2021.

---

> > ### Comment · Reviewer_g58u · 2023-11-22
> >
> > Thank you for your response. I will keep my score.

---

> > > ### Author Response · Authors · 2023-11-23
> > > **Thank you for the constructive review.**
> > >
> > > Thank you for the reply! We improved the quality of our manuscript significantly based on your invaluable input, and we sincerely appreciate your effort.

---

### Author Response · Authors · 2023-11-18

We once again thank all reviewers for their commitment and valuable feedback. The reviewers' comments provided great insights to improve the manuscript. We were delighted that the reviewers found our work **interesting** and relevant to the community (g58u), theoretically and experimentally **sound** (f3RM), and very **thorough**, including many important details (wuLi).

We summarize the major modifications as follows:

* (wuLi, f3RM) Experiment on CNNs with a stronger baseline (ResNet18 on ImageNet) [Appendix A.8.1, Table 5]
* (f3RM) Performance on Language Models (GPT-2 on WikiText-103) [Section 4.1, Table 1]
* (g58u) Discussion on generalization gap [Appendix A.8.2, Table 6]
* (f3RM, wuLi) Illustration of Eq. 8 and Algorithm 1 [Section 3.3, Eq.9]
* (f3RM) Discussion on overhead from SVD [Appendix A.6]

---

### Meta-Review · Area_Chair_UMBd · 2023-12-12

**Metareview:**

Although the paper had borderline scores, I think it makes a nice technical contribution to the area of low-rank compression of neural nets. As noted by reviewer wuLi, the ideas in the paper are strongly related to learning the ranks of the layers (for which a few approaches exist, see references in the review), so the authors should briefly discuss that work.

**Justification For Why Not Higher Score:**

Reviewers not that enthusiastic

**Justification For Why Not Lower Score:**

See metareview.

---

### Decision · Program_Chairs · 2024-01-16

Accept (poster)